# SASIEv.1: A framework for seasonal and multi-centennial Arctic sea ice emulation

Sian Megan Chilcott [1,2], Malte Meinshausen [1], and Dirk Notz [2]

[1]School of Earth and Atmospheric Science, The University of Melbourne, Parkville, Melbourne, Victoria, Australia, 3010
[2]Max-Planck-Institute for Meteorology, Universität Hamburg, Hamburg, Germany, 20146

**Correspondence:** Sian Megan Chilcott (smchilcott@student.unimelb.edu.au)

**Abstract.** The high computational expense of complex climate models and their tendency to underestimate observational records of Arctic sea ice sensitivity to anthropogenic forcers, challenge our ability to assess the magnitude of forcing that will cause Arctic sea ice loss to cross critical thresholds. To address these limitations, we develop a parameterisation framework for Arctic sea ice emulation, SASIEv.1, that is calibrated to the response of sea ice area to global warming in physically-based
5  CMIP6 models and constrained to observations. Our constrained framework reduces the remaining budget of $CO_2$ that can be emitted while preventing seasonally ice-free conditions from 821 $GtCO_2$ by CMIP6 multi-model ensemble estimates to 380 $GtCO_2$. This suggests that limiting global warming to 1.5°C is sufficient to prevent a seasonally ice-free Arctic Ocean, whereas 2°C proves insufficient. Our results also provide insight into the future of winter sea ice over a greater ensemble range than previously possible, pinpointing the emission threshold at which the ice pack detaches from land, after which the ice pack
10  rapidly disappears to year-round ice free conditions.

# 1 Introduction

Arctic sea ice forms a complex yet fundamental component of the Earth system and is a sensitive indicator of global climatic changes. Since the satellite record began in 1979, the September ice pack has declined by more than 13% per decade in response to anthropogenic greenhouse gas emissions (Serreze and Stroeve, 2015).

Global climate models (GCMs) used in the most recent (sixth) phase of the Coupled Model Intercomparison Project (CMIP), are the most comprehensive tools we have for predicting how Arctic sea ice will change in the future. Their projections unanimously show continued year-round reductions in Arctic sea ice throughout the $21^{st}$ century, with the majority of models projecting a seasonally ice-free Arctic Ocean within the next 15 to 50 years, commonly defined as the first time sea ice area (SIA) falls below 1 million $km^2$ in a given month (SIMIP Community, 2020). Though models agree ice loss will persist into the future, there are large uncertainties surrounding their projections (SIMIP Community, 2020). One conceptually convenient metric to measure changes in sea ice is the 'sea ice sensitivity', which is generally defined as the amount of sea ice area lost per degree of global warming. However, GCMs tend to simulate a lower sensitivity of Arctic summer sea ice loss than has been observed (Mahlstein and Knutti, 2012; Rosenblum and Eisenman, 2017; Niederdrenk and Notz, 2018; SIMIP Community, 2020). Models that do simulate present day rates of sea ice loss also simulate considerably higher global warming than observations suggest (Rosenblum and Eisenman, 2017). This is known as the 'hot model' problem and is used to describe models that project climate warming in response to $CO_2$ emissions that is much larger than other lines of evidence suggest (Hausfather et al., 2022). Recent observations report the Arctic Amplification, defined as the warming ratio between the Arctic and global temperature, over the satellite period has warmed much faster than in CMIP6 models which tend to simulate a relatively constant Arctic Amplification over the $21^{st}$ century (Chylek et al., 2022; Rantanen et al., 2022; Douville, 2023; Chylek et al., 2023; Hay et al., 2024).

The majority of CMIP6 models limit runs to 2100 as they are too computationally expensive to analyse large numbers of scenarios over multi-centennial timescales (Balaji et al., 2017). A few studies have utilised the limited number of extended simulations, to analyse the winter sea ice response to warming, using the limited subset of available models with extended simulations to 2300, with particular interest in the possibility of a rapid disappearance of winter sea ice (Armour et al., 2011; Hezel et al., 2014; Bathiany et al., 2016; DeRepentigny et al., 2020; Hankel and Tziperman, 2021, 2023). These studies found that the linear decline of Arctic sea ice with cumulative $CO_2$ emissions to ice-free conditions exhibited through the summer months breaks down in winter. Arctic sea ice declines at a much slower rate in winter before reaching a threshold at which the amount of sea ice loss per emitted ton of $CO_2$ rapidly increases. A recent study from Ritschel (2024) attributes the rapid loss of Arctic winter sea ice to the detachment of the ice pack from land. They find that the detachment of the ice pack from land is linked to the timing of geographic muting, a term introduced by Eisenman (2010) to describe the blocking of the ice pack's expansion through the growth season, due to the presence of land masses surrounding the Arctic Ocean. This mechanism explains the slow decline of winter sea ice initially, as the theoretical retreat of winter sea ice is masked by the coastline. This causes the area of the coast bound ice to change minimally. Once the temperature becomes sufficiently high to prevent the ice

pack expanding to the coastline, the ice pack has 'detached' from land and exhibits a higher sensitivity to warming (Ritschel, 2024). As the timing of rapid winter ice loss tends to occur after the end of the $21^{st}$ century, the lack of extended projections in CMIP6 models prevents a thorough analysis of winter ice loss.

This study therefore has three key aims. First, we build on and extend insights gained from comprehensive, higher complexity model runs to develop a parameterisation framework that emulates the CMIP6 response of seasonal Arctic sea ice area to global warming at a much smaller computational expense. In a second step we evaluate whether the calibration of our parameterisations to CMIP6 models between 1850 and 2100, capture the non-linearity of winter sea ice when extended to 2300. This validation provides insight into the framework's potential to emulate winter sea ice in models that lack projections beyond 2100. Finally, this study aims to investigate whether the sensitivity of sea ice loss to global warming can be constrained to match observations, through bias corrections to the global warming and Arctic Amplification trends. This study therefore aims to integrate various lines of evidence to efficiently emulate plausible, long-term projections of SIA under a range of scenarios. This approach does not intend to offer a model for "off-the-shelf" use, but a methodology by which we and other users can efficiently investigate seasonal and long-term Arctic sea ice trends using the parameterisations presented.

## 2 Framework Overview

The SASIEv.1 (Seasonal Arctic Sea Ice Emulator version 1) framework is separated into two stages which are each comprised of three steps (fig. 1). Stage one involves parameterisation development and calibration to CMIP6 data, while stage 2 constrains our CMIP6 informed parameterisations to observations. Within each stage, step i emulates the Arctic Amplification, converting the global annual mean temperature to the Arctic annual mean temperature (Sect. 4.2 and 4.2.1). Step ii emulates the Arctic seasonal temperature cycle which converts our emulated Arctic mean temperature to the Arctic seasonal temperature (Sect. 4.3) and is then input into the sea ice parameterisation in step iii (Sect. 4.4). In the second stage, we combine the MAGICC global-mean temperature ensemble (Sect. 3), with our probabilistic constraint on Arctic Amplification in step ii (Sect. 4.2.1), which we then pass through remaining steps to produce constrained, probabilistic projections of SIA to 2300. We use the MAT-LAB programming software version R2024b to run the SASIEv.1 framework, training the parameterisations on the CMIP6, MAGICC, RCMIP and observational datasets described in Sect. 3. Using this approach, the SASIEv.1 framework can be applied without recalibration across all models listed in Table S1, along with their associated Shared Socioeconomic Pathway (SSP) scenarios and ensemble members. The framework's parameterisations can also be calibrated to accommodate CMIP6 models beyond those considered in this study. Given that we emulate the sea ice response with parameterisations that smooth out year to year variability, our emulation framework therefore does not attempt to capture the year to year natural variability of Arctic sea ice, rather the long-term median Arctic sea ice response.

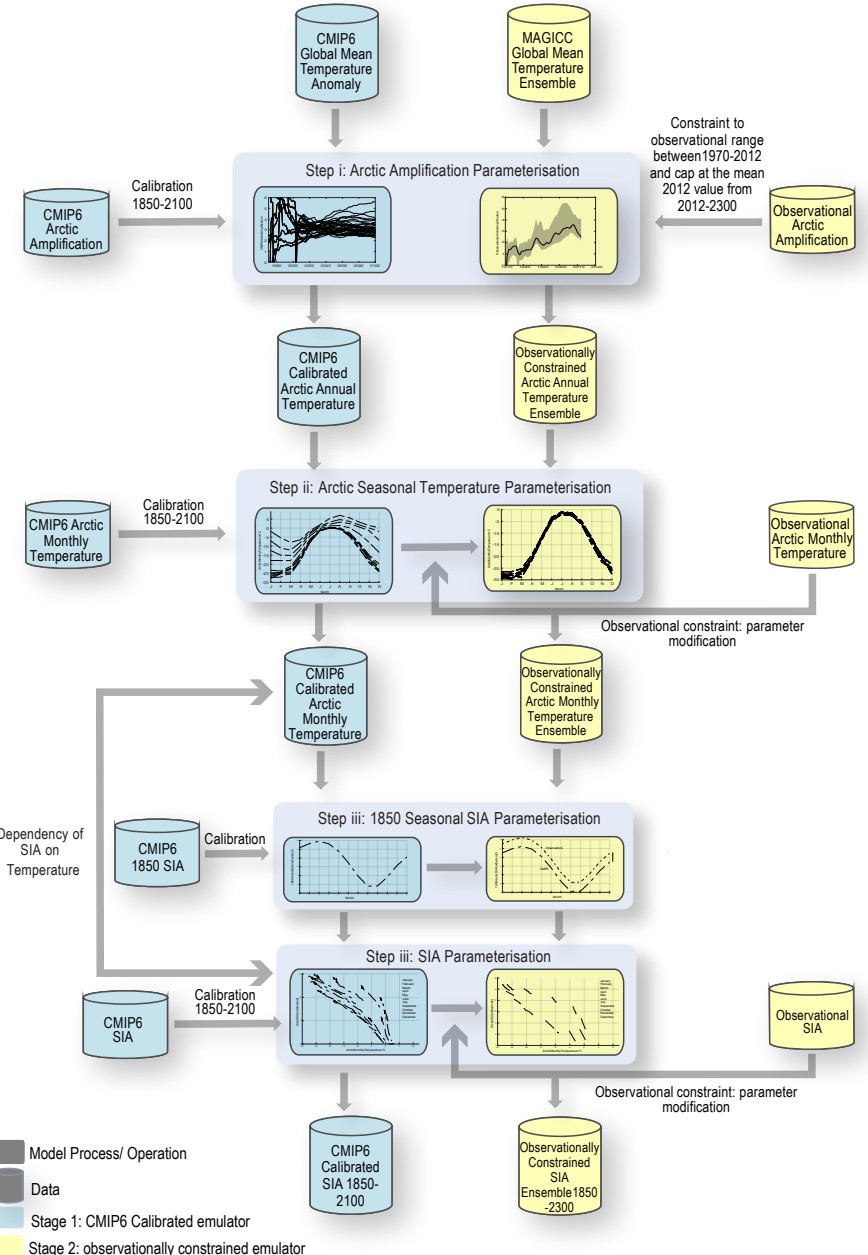

**Figure 1.** A work-flow of the parameterisation framework. Blue cylinders and boxes represent the calibration to CMIP6 models in Stage 1. Yellow cylinders and boxes represent the constraint to observational data in Stage 2. Cylinders represent the data that is used in each parameterisation step, while boxes represent the output from each parameterisation step. Refer to Sections 4.2 and 4.2.1 for more information on Step i. Refer to Section 4.3 for more information on Step ii. While section 4.4 provides more information on Step iii.

## 3    Data Collection and Processing

In the first stage, parameterisations have been calibrated over the period 1850-2100 against the corresponding first ensemble member of 13 selected CMIP6 models (see Table S1) for the SSP scenarios SSP5-8.5, SSP2-4.5 and SSP1-2.6 and their historical runs. We chose these scenarios and the calibration period based on the availability of both CMIP6 temperature and SIA projections, where at least five models also contained projections to 2300. These five models were selected for framework evaluation, presented in Sect. 5.1. By selecting scenarios that represent both the warmest and coolest projected futures, we ensure that our calibrated parameterisations encompass the full range of SIA responses to temperature. This enables the framework to be applied to intermediate scenarios not explicitly used in calibration, that lie within the bounds defined by the extremes. We also focus our calibration on the first ensemble member only, due to the small long-term variability between ensemble members, we instead focus our calibration on a range of models and scenarios. In the second stage, we use the MAGICC 600-member ensemble that has been constrained to represent the IPCC AR6 WG1 global warming projections and their uncertainty (Cross-Chapter Box 7.1, IPCC (2021) AR6 WG1; Nicholls et al. (2021)). The observational temperature datasets used to analyse the observed Arctic Amplification were the NASA's Goddard Institute for Space Studies Surface Temperature version 4 (GISTEMP) with a 1200km smoothing radius (Lenssen et al., 2019), the Berkeley Earth temperature dataset (BEST) (Rohde and Hausfather, 2020) and the Met Office Hadley Centre/Climatic Research Unit version 5.0.1.0 (HadCRUT5) dataset (Morice et al., 2021). The HADCRUT temperature range was obtained from the 200 member ensemble. In these datasets, near-surface air temperature is based on a combination of 2m temperature observations over land and sea surface temperature (SST) observations over the ocean. The global mean temperature anomaly is defined as the average temperature across the gridded dataset while the Arctic temperature anomaly counterpart is defined as the average of the area poleward of 65°N. Arctic absolute seasonal temperature observations are downloaded as a time series from the Berkeley product. The six observational northern hemispheric SIA datasets used include: the HadISST NSIDC monthly sea-ice area; HadISST original monthly sea-ice area; Comiso-Bootstrap monthly sea-ice area; NASA-Team monthly sea-ice area; OSI-SAF monthly sea-ice area and the Walsh monthly sea-ice area. Finally, cumulative $CO_2$ emission projections are taken from the historical fossil fuel, industrial and LULUCF (land use, land use change and forestry) datasets that were used for harmonising IPCC emissions scenarios from the AR6 WG3, RCMIP datasetes (IPCC, 2022; Nicholls et al., 2020). We acknowledge that different Earth System Models (ESMs) imply different LULUCF $CO_2$ emissions, as they often infer these internally from land-use patterns rather than using prescribed LULUCF emissions. To ensure consistency, we follow a common approach also used in the AR6 IPCC WG3, by using a harmonised LULUCF emissions time series based on historical bookkeeping estimates of anthropogenic emissions. While this introduces some uncertainty, it allows for a consistent estimate of cumulative emissions aligned with established carbon budget methods. Finally, as the cumulative $CO_2$ emission projections begin in 1750, to analyse the remaining carbon budget from 2023 we simply subtract the cumulative $CO_2$ emission in 2023 from the total carbon budget in each ensemble.

## 4 Framework Calibration

The calibration routine utilises the Nelder and Mead simplex optimisation method (Nelder and Mead, 1965; Lagarias et al., 1998), with a termination tolerance of $10^{-6}$ and a maximum iteration of 1000. We use the residual sum of the squared differences (RSS) for goodness-of-fit (GOF) diagnostics during the optimisation process. This setup iteratively evaluates the calibration factor values to ensure they result in an RSS global minimum between the CMIP6 data (with a running mean of 10 years applied) and the emulated data. The optimisation routine is initialised via a number of preliminary runs to generate a se-

ries of starting parameters, by randomly sampling values from a set of user-defined ranges for each parameter. Each calibrated parameter in the final model-specific set is designed to control all months in each model. Tables listing calibrated parameter values for all free variables can be found in the Supplementary (Table. S2 and Table. S3).

### 4.1 Parameterisation Framework

### 4.2 Step i: The Arctic Amplification Parameterisation

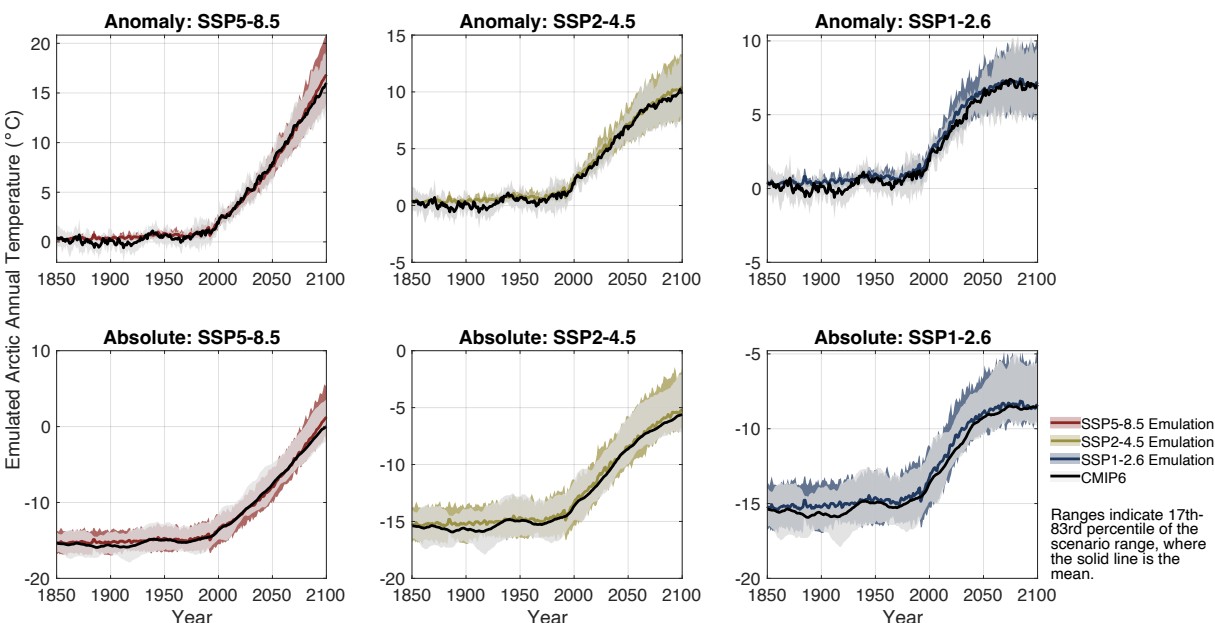

**Figure 2.** Emulation of CMIP6 calibrated Arctic annual mean temperatures. The top row represents the calibrated temperature anomaly, while the bottom row represents the calibrated absolute temperature. Red, yellow and blue shading represents the *likely* (17th-83rd percentile) calibrated model range for the scenarios SSP5-8.5, SSP2-4.5 and SSP1-2.6, with darker solid lines representing the mean in each scenario. Light grey shading represents the CMIP6 *likely* range and solid black lines represent the CMIP6 multi-model median.

In stage 1, we force our framework with the global mean temperature anomaly from each of the CMIP6 models used in this study. We split the emulation of the Arctic temperature into two parameterisations- annual (step i) and seasonal Arctic Amplification (step ii). Splitting these processes provides the basis for the observational constraint of annual Arctic Amplification in stage 2 of development.

We find that a simple linear regression between the global and Arctic annual mean temperature anomaly over the calibration period is sufficient to emulate the CMIP6 Arctic annual mean surface temperature (fig. 2):

$$tas_{(AAAB)} = \beta_{AA} \cdot \left(tas_{(GAB)} - tas_{(GREF)}\right) + tas_{(AREF)}, \tag{1}$$

where $tas_{(AAAB)}$ is the output Arctic annual absolute temperature, $tas_{(GAB)}$ is the global absolute temperature, $tas_{(GREF)}$ is the average 1850-1900 global absolute temperature and $tas_{(AREF)}$ is the average 1850-1900 Arctic absolute temperature, $(tas_{(GAB)}$ - $tas_{(GREF)})$ represents the global mean temperature anomaly and $\beta_{AA}$ is the regression coefficient representing the Arctic Amplification factor with a range between 2.5 and 4 about a median of 3, and is derived from the linear regression between the global mean and Arctic annual mean temperature anomaly. To generate the absolute temperature ($tas_{(AAAB)}$), the 1850-1900 mean Arctic absolute temperature ($tas_{(AREF)}$) for each CMIP6 model was added to the Arctic annual mean temperature anomaly ($\beta_{AA} \cdot (tas_{(GAB)} - tas_{(GREF)})$), to produce CMIP6 absolute temperatures.

### 4.2.1 Observational Constraint on Arctic Amplification

Given the disparity between simulated and observed Arctic Amplification trends, a more complex approach is required to constrain model trends to align with observational data. We note that due to the poor signal to noise ratio, we require warming to reach $0.45°C$ over a 20-year period before deriving Arctic Amplification from the observational record. As such, we focus our analysis on Arctic Amplification trends from 1970. The resultant parameterisation of our constraint on Arctic Amplification is as follows:

$$\text{Arctic Amplification}_{\text{r},\beta,\text{s}} = \frac{\beta_{\text{constantAA}}}{1 + \exp\left(S_{\text{linAA}} \cdot \left(-r_{\text{magicc}} + 0.5\right)\right)}. \tag{2}$$

Equation (2) is a sigmoid that initially assumes a linear increase of Arctic Amplification with global mean temperature at a rate dictated by $S_{\text{linAA}}$, which progresses asymptotically towards a constant Arctic Amplification prescribed by the parameter $\beta_{\text{constantAA}}$, after which the amplification factor will remain constant as warming continues. $\beta_{\text{constantAA}}$ represents randomly sampled values from the calibrated CMIP6 Arctic Amplification range ($\beta_{AA}$) in Eq. 1, while $S_{\text{linAA}}$ represents randomly

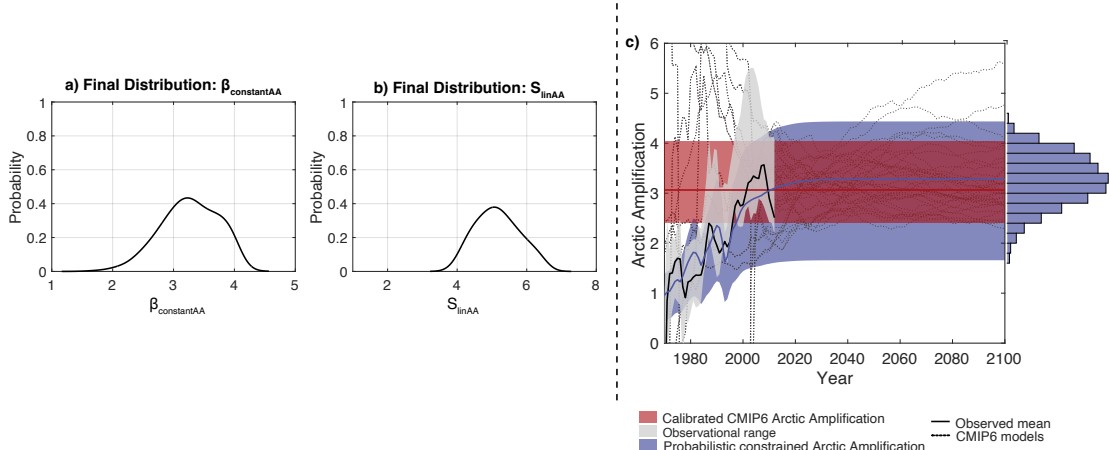

**Figure 3.** Constrained Arctic Amplification from our parameterisation framework, as presented in Eq. 2. a and b) represent the probability distributions of $\beta_{\text{constantAA}}$ and $S_{\text{linAA}}$ respectively, the driving parameters in our constrained Arctic Amplification equation (Eq. 2). c) shows the final constrained Arctic Amplification. Light grey shading represents the *'very likely'* (5th-95th percentile) observed Arctic Amplification range, whereas the black solid line is the observational mean. Red shading represents the *'very likely'* CMIP6 calibrated Arctic Amplification range (Eq. 1), whereas the red solid line is the CMIP6 calibrated multi-model mean. Blue shading represents the final constrained uncertainty range, while the blue histogram represents the frequency of each Arctic Amplification value in the range. We take 2012 as the final year of observations, as it is the last year a central point can be taken with 10 years either side to calculate the 21-year trend without reducing the number of years in the trend, which could bias the trend of the final years.

sampled values from the Arctic Amplification observational range (we provide further details in Supplementary Sect. S1.1). $r_{\text{magicc}}$ represents a randomly sampled MAGICC global mean temperature ensemble member, while its negative sign ($-r_{\text{magicc}}$) indicates that the sigmoid should increase rather than decrease with warming. We add a fixed factor of 0.5 to the exponential term to control the temperature at which the Arctic Amplification begins to increase, as the observed Arctic Amplification rises from a mean of approximately 1 as warming grades rise above 0.45°C (fig. 3). The division of $\beta_{\text{constantAA}}$ with the

non-linear exponential term in the denominator, $(1 + \exp(S_{\text{linAA}}(-r_{\text{magicc}}) + 0.5))$, controls the sigmoidal increase of Arctic Amplification with global warming at a rate prescribed by observations, while remaining constant thereafter to account for the modelled CMIP6 trend.

### 4.3    Step ii: The Arctic Seasonal Temperature Parameterisation

We develop our seasonal temperature parameterisation to reflect three 'key features' of warming on the evolution of the seasonal

temperature cycle. The first is the asymmetric warming between summer and winter which gradually reduces the amplitude of the seasonal temperature cycle over the calibration period (fig. 4), (Bintanja and Linden, 2013; Zhang et al., 2021; Screen et al., 2012). Secondly, the pre-industrial period (1850-1900) seasonal temperature cycle is relatively symmetrical as the rate of warming from winter to summer is similar in magnitude to the rate of cooling back to winter. However, as the Arctic warms and sea ice declines, the temperature cycle widens at its peak as the melt season lengthens and warm summer temperatures

persist into the autumn months. Finally, we find the temperature amplification is not linear in all months. From the mid-$21^{st}$

century the temperature amplification increases during the summer months (June-August), declines in autumn and early winter

(September-January), and remains relatively constant through late winter and spring (February-April) (Supplementary fig. S2).

The likely cause of the seasonal amplification difference is outlined by Dai et al. (2019), and is related to the seasonal exchange

of fluxes between the ocean surface and atmosphere, which respond differently to variations in the rate of sea ice loss to rising

$CO_2$ emissions. We represent these trends via a nested, exponential cosine function that emulates the shape of the seasonal

temperature curve in each year, and is controlled by the Arctic annual mean temperature:

$$P = F_{\text{amp}} \cdot \cos(m \cdot G_{\text{waveLen}} - E_{\text{pShift}} \cdot \exp\left(\cos(m^{A_{\text{meltLen}}})\right)), \tag{3a}$$

$$tas_{(AMAB)} = P(1 + h_{\text{vertShift}}), \tag{3b}$$

where $tas_{(AMAB)}$ is the seasonal Arctic mean temperature in degrees Celsius emulated in step i, $m$ is an equally spaced

value between 0 and $2\pi$ representing the reference point of each month of the year (0 and $2\pi$ represent January of year $t$

and January of year $t+1$ respectively) and $F_{\text{amp}}$, $G_{\text{waveLen}}$, and $A_{\text{meltLen}}$ are model-specific calibration parameters dependent

on the Arctic annual mean temperature. $F_{\text{amp}}$ represents the amplitude, $G_{\text{waveLen}}$ controls the wavelength, while $A_{\text{meltLen}}$ (Eq.

3c) is a simple cosine function that controls the change from a basic cosine curve to a shallow, linear temperature decline,

as warmer summer temperatures gradually pervade into September, October, November and December (Supplementary fig.

S3). $E_{\text{pShift}}$ is a non-temperature dependent and non-model specific parameter with a value of 0.3, that fixes the phase shift of

the curve and therefore controls the rate of change from summer to winter temperatures. Finally, $h_{\text{vertShift}}$ (Eq. 3d) is a non-

optimised, dimensionless function that modulates '$P$' by offsetting the temperature to ensure the emulated Arctic annual mean

temperature input at time $t$ is equal to the mean of the parameterised monthly temperature curve and is calculated at every time

step. $F_{\text{amp}}$ and $G_{\text{waveLen}}$ are calculated from the calibration coefficients $f_1$, $f_2$, $g_1$ and $g_2$ that are optimised through a series

of simple linear regressions dependent on the Arctic annual mean temperature; ($F_{\text{amp}} = (f_1 \cdot tas_{(AAAB)}) + f_2$) and ($G_{\text{waveLen}} =$

$(g_1 \cdot tas_{(AAAB)}) + g_2$), Supplementary table. S2. We acknowledge that while $F_{\text{amp}}$ and $G_{\text{waveLen}}$ are related, the complexity of

the system did not allow us to find a functional form to capture their relationship. As such, we handle these two parameters

separately.

$$A_{\text{meltLen}} = \cos((tas_{(AAAB)} \cdot -a_1) - a_2) + a_3, \tag{3c}$$

$$h_{\text{vertShift}} = \frac{tas_{(AAAB)} - \overline{P}}{F_{\text{amp}}}. \tag{3d}$$

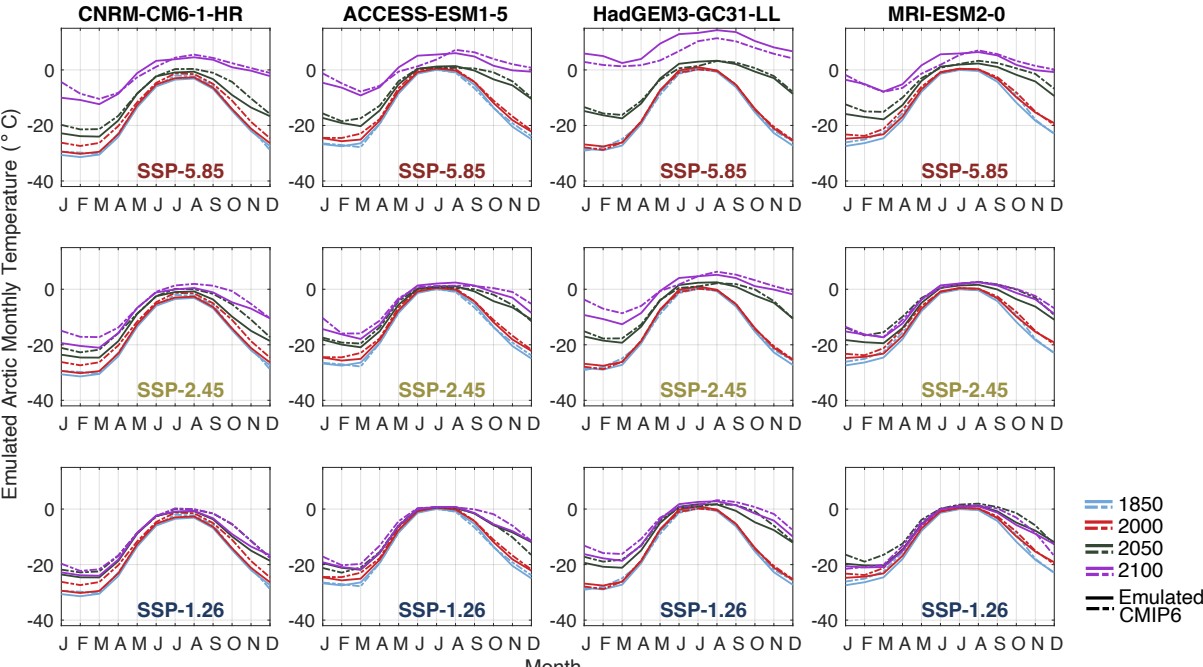

**Figure 4.** Emulation of the CMIP6 Arctic seasonal temperature cycle in 4 selected years and models. Each column represents our emulation of one model, where the top, middle and bottom rows represent SSP5-8.5, SSP2-4.5 and SSP1-2.6 respectively. Solid lines represent our emulation of a selected CMIP6 model and dashed lines of the same colour represent the CMIP6 data. Only emulation of the years 1850, 2000, 2050 and 2100 are displayed for visualisation purposes, however calibration was conducted over each year between 1850 and 2100, were one year represents one curve.

The first key feature (see above) is encapsulated by parameter $F_{\text{amp}}$, which reduces the amplitude of the curve with warming. The second is satisfied by the exponent of the cosine $\exp\left(\cos(m^{A_{\text{meltLen}}})\right)$. This feature lengthens the summer season as the Arctic annual mean temperature increases. Although our function initially assumes a basic cosine curve at lower values of $tas_{(AAAB)}$, this feature creates the increasingly asymmetric and broad peaked curve as $tas_{(AAAB)}$ rises, generating a left skewed curve at higher Arctic annual mean temperatures. Finally, adding the exponent into the nested cosine $\left(\cos(m(\cos(m^{A_{\text{meltLen}}})))\right)$ prevents the temperature in the autumn months rising as fast as the winter months, reducing the amplification from the mid-21$^{st}$ century while simultaneously increasing the amplification during summer. The combined decline in amplitude ($F_{\text{amp}}$) and increase in wavelength ($G_{\text{waveLen}}$) prevents the increase in summer amplification initially, however as the wavelength increases

the curve becomes more left skewed causing the rate of July and August warming to increase. This satisfies the third key feature of the seasonal temperature evolution we identify in CMIP6 models.

Analysis of the few models with runs to 2300 (CanESM5), show the shape of the temperature curve doesn't evolve significantly past 2100. As such, we cap the evolution of our calibration parameters to ensure they remain constant past an Arctic annual mean temperature of 6°C (Supplementary fig. S4). Past this temperature, $h_{\text{vertShift}}$ (Eq. 3d) is the only parameter that continues

to change. We find the temperature parameterisation produces a plausible timeseries to 2300 using this method (Supplementary fig. S5).

### 4.3.1   Bias Corrections to Arctic Monthly Temperature Parameterisation

CMIP6 projections of the mean 1979-2020 Arctic summer (April-July) temperatures are on average 2°C warmer than observed (Supplementary fig. S6). To overcome this, we apply a constant offset to our CMIP6 calibrated temperature in each month,

which we define as the difference between the 1979-2020 mean from observations and that of our emulation. We acknowledge the 1979-2020 period is a short time frame and internal variability could be a large factor causing this difference. However, as future observations become available and the impact of internal variability is better understood, this bias correction can be updated in future versions of the SASIE framework.

CMIP6 models also project a weaker summer (July and August) warming trend than is observed (Supplementary fig. S6).

Simulated July and August temperatures increase slowly from 1980 to 2050 before rising significantly thereafter, whereas observed temperatures rise rapidly from ∼1980. We address this by forcing the calibration parameter '$F_{\text{amp}}$', which represents the amplitude of the temperature curve in each year in Eq. (3b), to remain constant until the annual mean Arctic temperature reaches an absolute level of −8°C. Our bias correction ensures the summer temperatures increase at the observed rate, while protecting our emulation of the lengthening melt season (Serreze and Barry, 2011).

**4.4   Step iii: The SIA Parameterisation**

We parameterise the response of Arctic sea ice area to the seasonal Arctic warming trend, as this is ultimately the temperature sea ice responds to (as opposed to global-average warming itself). As CMIP6 models reproduce the observed Arctic warming trend well, this method also ensures our SIA projections respond to Arctic warming trends that match the observed, reducing the number of bias corrections in stage 2.

**4.4.1   The Seasonal Melt and Growth Weighting Scheme**

The response of Arctic sea ice to temperature is impacted by many different physical processes such as snow cover and ocean heat content changes. Variations in sea ice thickness in particular offer insight into how the different rate of ice area growth

and melt can be incorporated into our framework. During autumn freeze-up, a thin layer of sea ice will rapidly form over the Arctic Ocean. However, during the melt season, a greater thickness of ice needs to melt before the SIA is greatly affected. This

produces a differing inertia to temperature changes between the growth and melt of SIA, where for a given temperature change the loss or gain of SIA will be greater in autumn compared to that of spring (Ritschel, 2024; Eisenman, 2010). We therefore apply a weighting scheme to the emulated temperature that forces our SIA parameterisation, to account for the inertia in the sea ice system (Supplementary fig. S7). The updated temperature ('$tas$') therefore becomes a product of the current month, $tas_{(m)}$ and the previous month, and is computed via the following weighting scheme:

$$tas_{(new)} = \frac{(tas_{(m)} \cdot w_m) + (tas_{(m-1)} \cdot 1 - w_m)}{w_m + 1 - w_m}, \tag{4a}$$

where $tas_{(m)}$ is the Arctic temperature at the monthly timestep $m$, $tas_{(m-1)}$ is the temperature from the previous month and $w_m$ and $1 - w_m$ are the weightings applied to the current and previous month respectively.

When applying a greater weight to the previous month's temperature in spring, the updated temperature is colder as a greater weight is placed on the colder winter temperatures (Supplementary fig. S7b). Spring SIA is therefore forced with a colder

temperature to express the slower decline of SIA for a given temperature change through the melt season. Whereas, the updated temperature is higher in autumn as it becomes a function of the the previous warm summer month, to capture the faster growth of sea ice for an equivalent temperature change through the growth season.

Due to the effect of 'geographic muting' described in Sect. 1, Arctic summer sea ice declines linearly with Arctic warming while the loss of winter sea ice accelerates with warming as the ice pack detaches from land (Ritschel, 2024). Given these

235 characteristics, we develop the following parameterisation to reflect the sigmoidal decline of SIA with Arctic warming:

$$SIA = \frac{(SIA_{max} + d_{offset}) \cdot S_{fix} - L_{linSIA}(tas - tas_{(t=0)})}{(1 + \exp{(tas - b_{nonlinSIA})})}, \tag{4b}$$

$$S_{fix} = (1 + \exp{(tas_{(t=0)} - b_{nonlinSIA})}), \tag{4c}$$

where $SIA_{max}$ (Eq. 4d) is the seasonal SIA in 1850 and is optimised for each model, $L_{linSIA}$, $b_{nonlinSIA}$ and $d_{offset}$ are model dependent calibration factors and '$tas$' is the weighted Arctic seasonal temperature emulated in step ii.

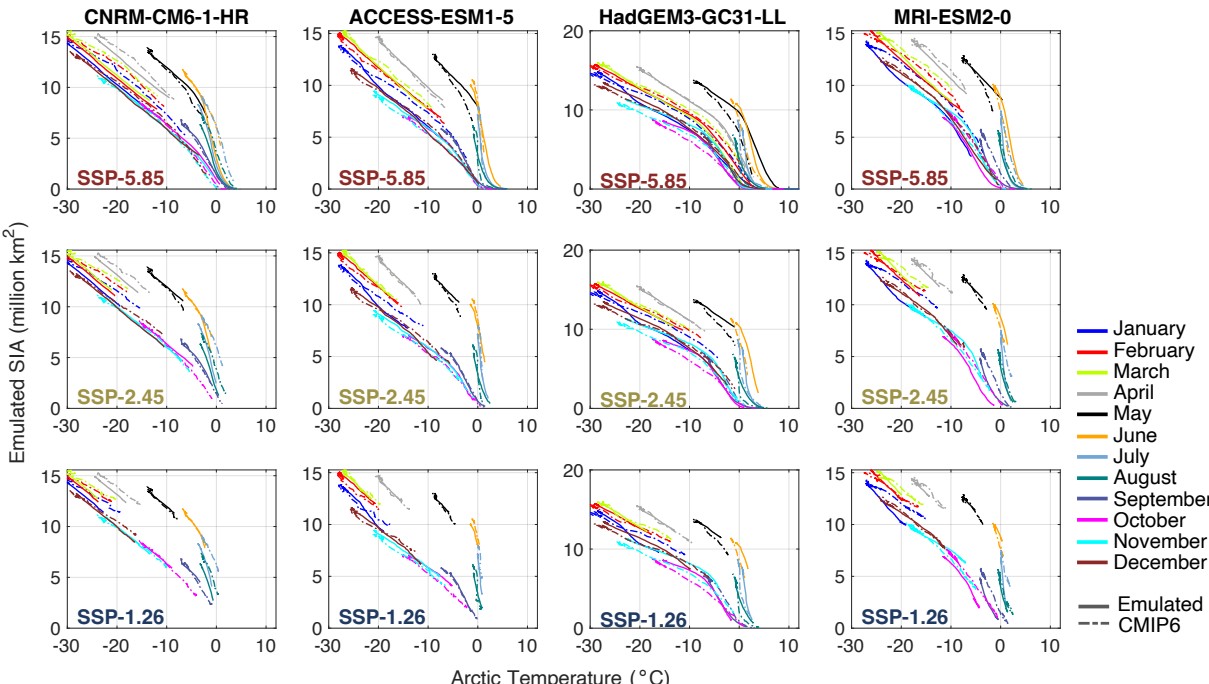

**Figure 5.** Emulation of the CMIP6 Arctic seasonal Sea Ice Area cycle in 4 selected models. Each column represents our emulation of the SIA under three scenarios from one CMIP6 model, where the top, middle and bottom rows represent SSP5-8.5, SSP2-4.5 and SSP1-2.6 respectively. Solid lines represent our emulation of the CMIP6 SIA in each month, and dashed lines of the same colour represent the CMIP6 data.

The resultant functional form initially assumes a slow linear decline of SIA with temperature, which progresses asymptotically towards a constant SIA of 0 million km$^2$ at higher temperatures (fig. 5). Our SIA parameterisation is therefore derived from the division of a linear and non-linear ('$tas$') term. The linear response term on the numerator $L_{\mathrm{linSIA}}(tas - tas_{t=0})$ controls the initial linear decline of SIA with temperature and is scaled via the subtraction of the '$tas$' in 1850 from subsequent '$tas$' in each year; while the exponential '$tas$' term $(1 + \exp(tas - b_{\mathrm{nonlinSIA}}))$ on the denominator, controls the sigmoidal decline of SIA and

therefore the increase in the sensitivity of sea ice loss to warming in the colder months. The starting term $(\mathrm{SIA}_{\mathrm{max}} + d_{\mathrm{offset}}) \cdot S_{\mathrm{fix}}$ fixes the initial SIA in each month, and is the maximum SIA susceptible to melting. $\mathrm{SIA}_{\mathrm{max}}$ is offset via $d_{\mathrm{offset}}$ to account for the chaotic nature of pre-industrial sea ice fluctuations for small warming grades, and the imperfect nature of emulations. An exponential '$tas$' term is added for the timestep 0 to account for the increase in the exponent of '$tas$' as the initial temperature nears 0°C in the summer months, ensuring that the maximum SIA is equal to $\mathrm{SIA}_{\mathrm{max}} + d_{\mathrm{offset}}$ in the summer months. The

CMIP6 seasonal cycle of SIA in 1850 contains a few key features that justify the more complex adapted sine function in Eq. (4d) (Supplementary fig. S8) below:

$$\text{SIA}_{\text{max}} = A_{\text{amp}} \cdot \left( -\exp\left( \sin\left( m^{C_{\text{crest}}} \cdot W_{\text{waveLen}} - K_{\text{pShift}} \right) \right) \right) + V_{\text{verShift}}, \tag{4d}$$

$$K_{\text{pshift}} = \left( W_{\text{waveLen}} \cdot 0.8003 \right) + 1.5016, \tag{4e}$$

where $m$ is again an equally spaced value between 0 and $2\pi$ representing the reference point of each month of the year (0 and $2\pi$ represent January of year $t$ and January of year $t+1$ respectively). Calibration factors $W_{\text{waveLen}}$, $A_{\text{amp}}$ and $V_{\text{vertShift}}$ control the wavelength, amplitude and vertical shift respectively. The parameter $K_{\text{pshift}}$ controls the phase shift and is a constant derived from a linear regression of $w_{\text{waveLen}}$. The exponent of the sine $-\exp\left(\sin(m^{C_{\text{crest}}})\right)$ is added to flatten the trough of the curve, as August and September have similar pre-industrial values. Whereas the $m^{C_{\text{crest}}}$ term modulates the roundness of the peaks,

to ensure the SIA increases sharply from January to its peak in March. This term allows the wavelength to capture the slow transition from winter SIA to summer SIA, while capturing the faster transition from summer SIA to winter SIA in 1850. Finally, we use a negative exponent to represent the decline in SIA through the summer months. The parameter $K_{\text{pshift}}$ then adds a small phase shift to adjust the phase to each specific model. The $\text{SIA}_{\text{max}}$ factor is first optimised and then input into the SIA calibration.

### 4.4.2   Bias Corrections to the Sea Ice Area Parameterisation

Similarly to the Arctic seasonal temperature bias correction, we find the modelled 1979-2014 mean SIA between June and September is on average 1.14 million km$^2$ smaller than observations suggest (fig. S6). To address this, we adjust $\text{SIA}_{\text{max}}$ in each month by subtracting the difference between the 1979-2020 mean observed SIA and our emulated mean.

We also find that applying the bias-corrected temperature from Sect. 4.3.1 to our SIA parameterisation generates a much larger

sea ice loss per degree of warming than is indicated by both CMIP6 models and observations (Supplementary fig. S9). This is to be expected as our corrected temperatures rise at a faster rate between 1979 and 2050 than in CMIP6 projections, while our calibrations representing the response of SIA to Arctic warming remain unchanged. This causes the sensitivity of SIA loss to increase at a cooler temperature, as the sensitivity parameter '$b_{\text{nonlinSIA}}$' (Eq. 4b), is reached much earlier than in our calibration. To overcome this, we adjust '$b_{\text{nonlinSIA}}$' by subtracting the difference in the mean 1979-2050 Arctic temperature produced from

bias correcting the calibration parameter '$F_{\text{amp}}$', and the CMIP6 calibrated temperature where the calibration parameter '$F_{\text{amp}}$' has not been bias corrected. This approach forces the sensitivity to increase at the calibrated temperature and year that CMIP6 models suggest, while also allowing the temperatures to rise at the same rate as observations. The impact of this difference is minimal in the colder months but more pronounced in the summer, as '$b_{\text{nonlinSIA}}$' tends to be reached before 2050 through the summer months while it is reached much later in winter.

## 5   Results

### 5.1   Evaluating the CMIP6 Framework's Performance in Capturing Rapid Ice Loss

Through this section, we evaluate the ability of our CMIP6 calibrated parameterisations to reproduce the CMIP6 response of SIA to warming in each month when extended to 2300, with particular focus on its ability to project the rapid disappearance of March sea ice. This approach aligns with our aim to emulate long-term winter sea ice loss using only information available within the $19^{th}$, $20^{th}$ and $21^{st}$ centuries. This provides insight into the framework's potential to emulate winter sea ice in models that lack projections beyond 2100. While multiple evaluation strategies are possible, we focus on this method as it directly tests the long-term generalisability of our parameterisation framework. Two additional evaluation approaches are provided in the Supplementary information (fig. S11 and fig. S12) to provide an evaluation of our framework on a non-calibrated scenario and ensemble member. However here, we focus on the performance of long-term sea ice reproduction. We note that our use of the term '2100 calibration' refers to our calibration between 1850 and 2100, while our 'out of sample' test refers to output produced by forcing this calibration with extended global warming projections to 2300.

We find our 2100 calibrations hold 'out of sample', projecting an ice-loss between 1850 and 2300 that aligns with the CMIP6 trend for each month and model, successfully capturing the rapid loss of March sea ice to both global and Arctic warming (fig. 6 and Supplementary fig. S13). We evaluate the goodness of fit (GOF) between our emulation of SIA to 2300 with the corresponding CMIP6 model via the root mean squared error (RMSE), dividing by the number of calibrated model years. The GOF statistics of the March sensitivity to Arctic warming show an RMSE of 0.0037 and a correlation coefficient of 0.99, while the fit against global warming produces a RMSE of 0.07. Both statistics suggest the successful performance of our parameterisations to 2300. We propose the slightly weaker fit to global warming is because this relationship carries the uncertainties from all three steps of our framework, while the relationship to Arctic warming only carries the uncertainties from one step. While the GOF statistics suggest a successful fit in most months, May in IPSL-CM6A-LR shows a slightly weaker fit (fig. 6). While our weighting scheme accounts for the lag in the response of sea ice to temperature between the growth and melt seasons, May technically requires a larger weight $(w_m)$ than the growth months. However, our parameterisation only uses a single weight across all SSPs and timescales (1850–2100) to reduce the number of free parameters, missing this variation. Despite a weaker melt-season fit, it remains sufficient for emulation.

Our results suggest that our calibration to 2100 is sufficient to project the non-linearity of winter sea ice loss, however, the majority of models analysed indicate that rapid March ice loss occurs after 2100. We attribute this ability of our parameterisations to project the rapid March ice loss, despite it occurring outside the calibration range of our data, to the combination of the sigmoidal nature of our SIA parameterisation (Eq. 4b), which inherently assumes the rate of ice loss will increase once the threshold temperature controlled by '$b_{\mathrm{nonlinSIA}}$' is reached, and the temperature weighting scheme (Eq. 4a). Although March generally does not warm enough to cause an increase in the rate of ice loss before 2100, the early winter and melt season months (November, December and May), show an increase in the sensitivity before 2100. While the specific temperature 'threshold'

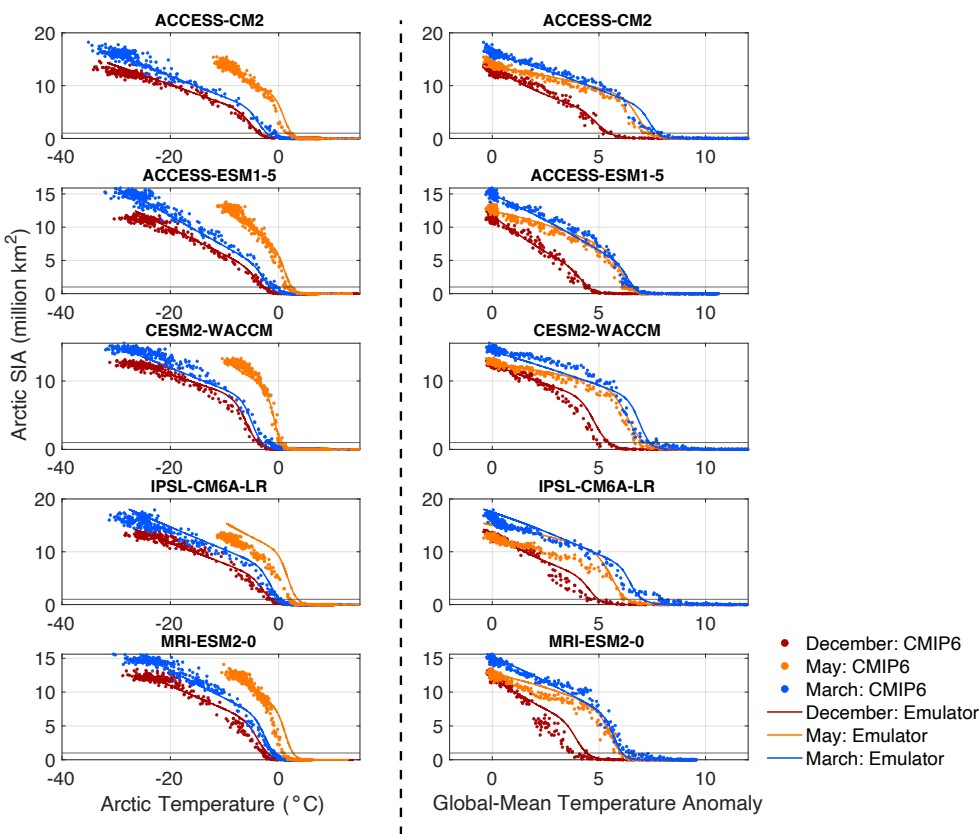

**Figure 6.** Comparison of the winter sea ice response to Arctic and global warming trends in 'out of sample' extended runs between our emulation framework and CMIP6 data. The left column shows the SIA response to Arctic warming in extended projections between 1850 and 2300, while the right panel shows the SIA response to global warming in extended projections over the same period. Each row represents one of the five CMIP6 models analysed for each warming trend. Solid lines represent our emulation of the CMIP6 response while scatter points represent the CMIP6 model response. We only show three months here for clarity.

at which the sensitivity increases at varies between the months and models, it is our calibration of the sensitivity in the early winter and melt season months that informs ice loss after 2100.

The Arctic temperature at which rapid ice loss occurs is relatively similar in November and December, whereas the threshold
temperature in May is much warmer (Supplementary fig. S7). We attribute this to the different paces in growth and melt of Arctic sea ice, discussed in Sect. 4.4.1, suggesting the temperature threshold at which the ice pack detaches from land, causing rapid ice loss, will be cooler in the growth season than in the melt season (Ritschel, 2024). Our temperature weighting scheme presented in Sect. 4.4.1, (Eq. 4a), parameterises this difference which subsequently updates the threshold temperature of rapid ice loss. Without the weighting scheme, the sigmoidal nature of our parameterisation would cause the SIA to decline at the same

temperature in each month, at the value set by the sensitivity parameter '$b_{nonlinSIA}$' (Supplementary fig. S7a). By calibrating to the rapid ice loss in the early growth (November and December) and melt season (May) months that exhibit rapid ice loss before 2100, our calibration scheme understands the different temperature required to melt and grow ice in both seasons and generates a single value for the weight and sensitivity parameters ($w_m$ and $b_{nonlinSIA}$) to capture the seasonal cycle in each model. Our SIA parameterisation then assumes that $w_m$ and $b_{nonlinSIA}$ will have the same effect on the months that don't exhibit rapid ice loss before 2100 as the months that do over the calibrated period.

Through this section, we have evaluated the performance of the SASIEv.1 framework by assessing its ability to capture the abrupt loss of winter sea ice, outside of the calibration period. Two additional evaluation approaches are also provided in the Supplementary information (fig. S11. Here, we first applied the calibration parameters from the first ensemble member of each emulation model to the non-calibrated scenario SSP4-6.0, to consider the out-of-sample performance. We additionally applied these parameters to the full suite of ensemble members from a test case model, CanESM5. We find our parameterisations reproduce SSP4-6.0 and the majority of ensemble members tested well, indicating that our approach provides a good approximation of the ensemble range and can be applied to scenarios beyond those calibrated. While a large aim of framework development involved capturing the rapid loss of winter sea ice, the framework is also designed for broader application. Future work could explore its performance across a wider range of experiments, such as overshoot pathways where we expect our SIA parameterisation to respond to temperature recovery, as seen in SSP1-2.6 (Supplementary Fig. S14). Assessing its ability to emulate experiments beyond the temporal and structural assumptions of the calibrated ESMs would be valuable for future framework application.

### 5.2    Evaluating the Framework's Performance in Capturing the Observed Sensitivity

Here we evaluate whether the constraints we apply to our parameterisations in stage 2, are sufficient to capture the observed sensitivity of sea ice loss to global warming. To do so, we compare our emulator to the 'plausible' range defined by the SIMIP Community (2020), which accounts for both the structural and internal variability, by taking two standard deviations of the multi-model CMIP6 ensemble as the uncertainty range around the observed sensitivity of sea ice loss between 1979 and 2014. We take the sensitivity as the regression coefficient between the global temperature and SIA projections over the period 1979-2014 (fig. 7). Although our emulator generates projections in all months, we focus on the September sensitivity as CMIP6 models tend to underestimate the summer sensitivity while reproducing the winter trends relatively well.

When we constrain our parameterisations to address both the 'hot model' problem and the Arctic Amplification bias, we project a sensitivity that matches the 'plausible' range relatively well in most months. In September, our framework projects that a median of $-4.06$ million km$^2$ of sea ice is lost per degree of warming between 1979 and 2014, aligning with the 'plausible' September sensitivity of $-3.95$ million km$^2$/°C relatively well. Similarly, when assessing how well our projections aligns with the 'plausible' amount of sea ice loss per metric ton of emitted $CO_2$, we find our observationally constrained emulator projects a sensitivity of $-2.5 \pm 0.95$m$^2$/t$CO_2$, which falls within the 'plausible' estimate of $-2.73 \pm 1.37$m$^2$/t$CO_2$. While

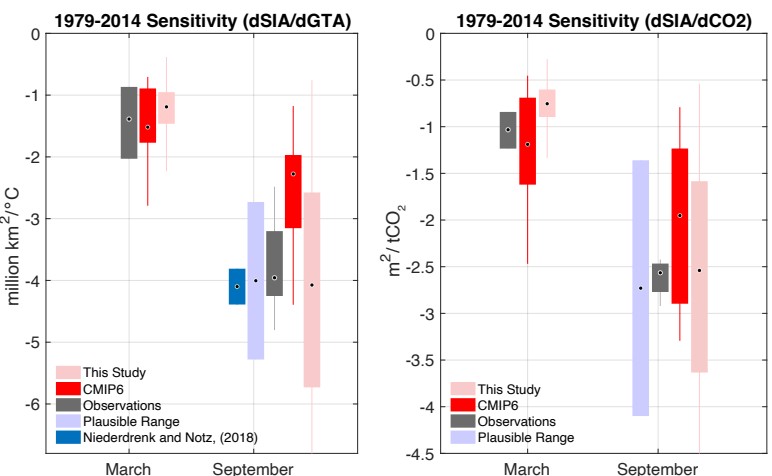

**Figure 7.** Comparison of the SIA sensitivity to global warming and cumulative $CO_2$ emissions from 1850 in March and September between our framework output, CMIP6 models and observations. For each month, dark red boxplots denote the sensitivity in CMIP6 models, dark grey boxplots denote the sensitivity from observations, light blue boxes represent the 'plausible' range from observations, light pink boxplots show the sensitivity generated from our observationally constrained emulator, dark blue boxes represent the sensitivity from Niederdrenk and Notz, (2018). All boxplots extend from the lower to upper quartile values over the interquartile range, with a point at the median. Whiskers show the full range of values, excluding outliers.

our interquartile range is slightly larger than the 'plausible' against global warming, and slightly smaller than the 'plausible' against $CO_2$ emissions, this is expected due to the probabilistic nature of our framework constraints and the influence of model uncertainty from CMIP6 multi-model calibration.

## 5.3 Application: What is the Temperature and Remaining Carbon Budget Necessary to Prevent Arctic Sea Ice Loss Crossing Critical Thresholds?

Through the previous sections we have evidenced that our SASIEv.1 parameterisation framework can reproduce observed sensitivity trends, while also successfully extending projections to 2300. Here we apply the framework to understand the impact of its constraints and extension on key questions within the sea ice discourse. We separate our analysis into two regimes; a linear and non-linear regime. The linear decline of sea ice with cumulative $CO_2$ emissions- measured from 1850- in July through to December, which we define as the linear regime, provides the basis for the calculation of a finite remaining carbon budget, alongside an assessment of our current progress towards IPCC warming targets. In a second step, we analyse the non-linear ice loss exhibited in extended projections between January and June, which we define as the non-linear regime, to pinpoint the temperature and $CO_2$ emission threshold at which winter sea ice detaches from land (Supplementary fig. S15). While previous studies have assessed the global mean temperature threshold at which rapid winter ice loss occurs, to the best

of our knowledge this is the first study to pinpoint the $CO_2$ emission at which this occurs, over a much wider ensemble range than previously possible. This approach also highlights the flexibility of our framework to investigate sea ice change beyond temperature comparisons, and to situate results within broader climate mitigation benchmarks.

## 5.4 Linear Regime

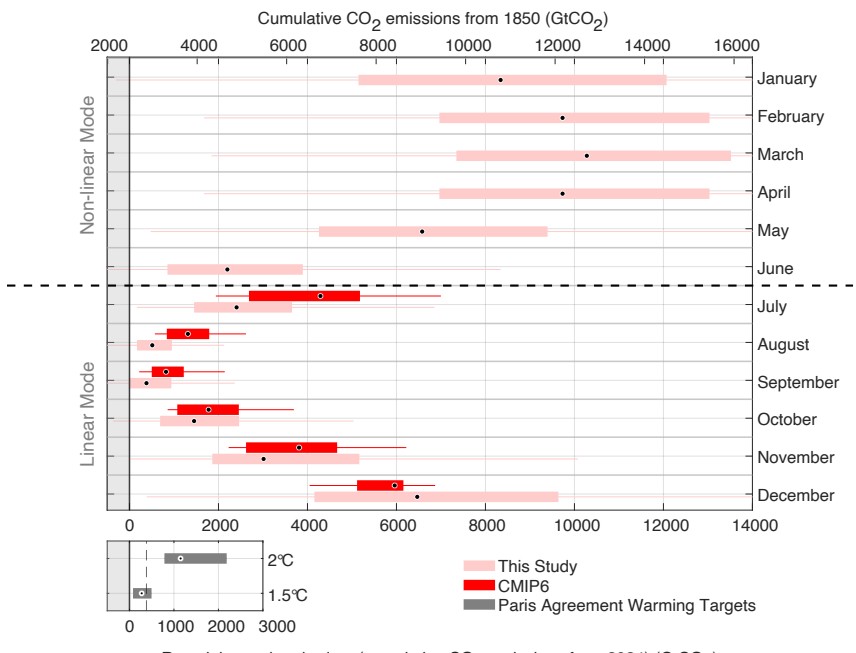

**Figure 8.** The carbon budget to prevent a seasonally ice-free Arctic Ocean in the linear mode months (July-December) and the carbon budget to prevent rapid ice loss occurring in the non-linear mode months (January-June). Light pink boxplots represent the remaining carbon budgets from 2024, calculated from our emulator. Red boxplots show the remaining carbon budget calculted from the CMIP6 models used in this study. Grey boxplots in the bottom panel compare our results with the remaining carbon budget to prevent Paris Agreement warming targets. The dashed line in the bottom panel compares our emulator's budget for a 50% chance of preventing ice-free conditions in September with the remaining carbon budget to prevent a 1.5°C warming target. Finally, grey vertical shading represents carbon emissions that would exceed the budget to prevent an ice-free ocean or to prevent non-linear ice loss in the winter months. Boxplots represent the interquartile range of the data with the median shown through the black and white dot.

When translating the constrained sensitivity from the emulation framework into the remaining carbon budget, we estimate that there is a 50% chance Arctic September sea ice will be lost for an additional 380 Gigatonnes (Gt) of $CO_2$ emissions from 2024 (fig. 8) (IQR: $-14$ $GtCO_2$ and 940 $GtCO_2$). Our estimate is smaller than CMIP6 projections, which suggest a median of 821 $GtCO_2$ (IQR: 500 $GtCO_2$ and 1220 $GtCO_2$) will cause a seasonally ice-free Arctic Ocean. If we assume the average current

emission trends of 38 Gt of $CO_2$ per year will continue into the future (Lamboll et al., 2023), our updated limit of 380 $GtCO_2$ will be reached within the next decade.

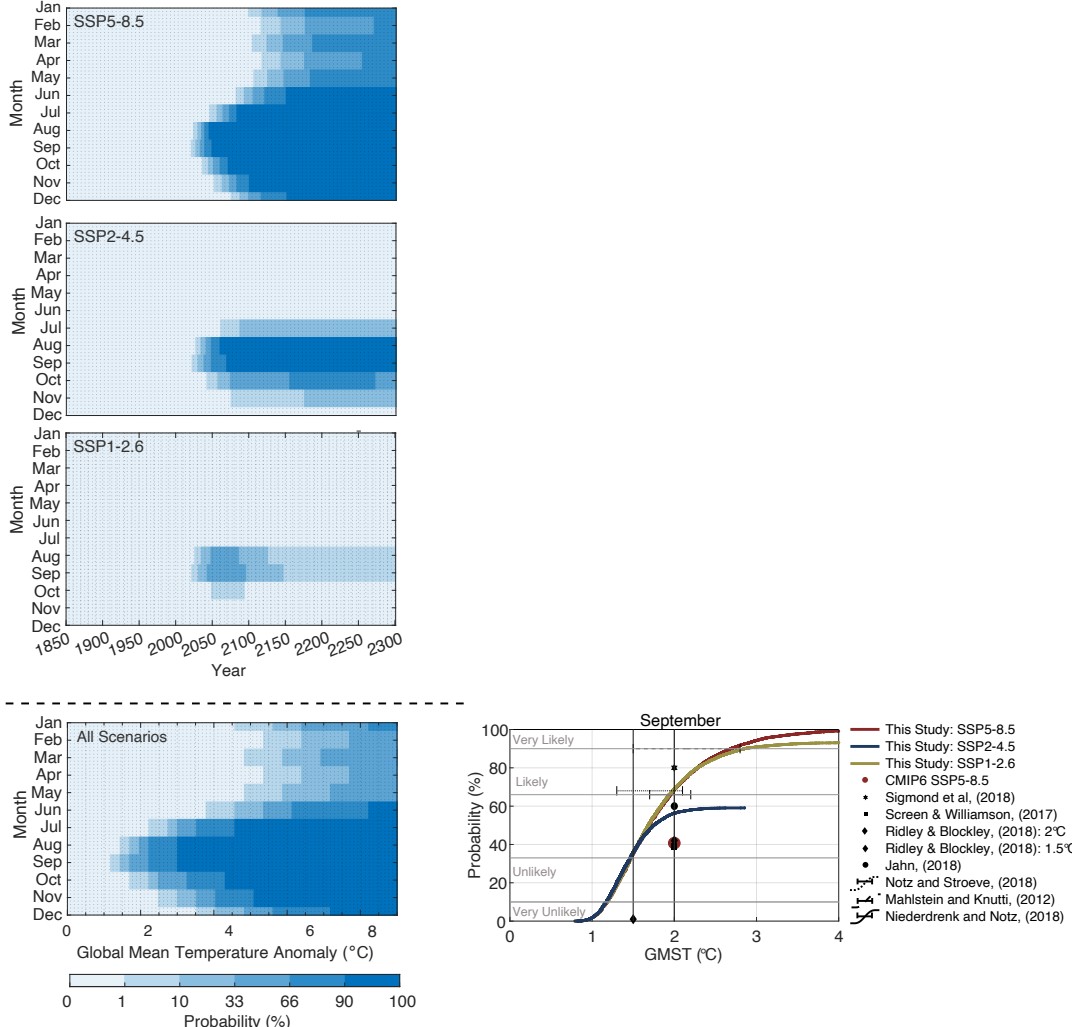

**Figure 9.** The probability of ice-free conditions in each month, year and global mean temperature by 2300 generated from our observationally constrained parameterisations. The probability in each year under SSP5-8.5, SSP2-4.5 and SSP1-2.6 are given in the upper three panels. The probability as a function of global temperature is shown in the bottom panel, and is calculated from the likelihood of ice-free conditions at each global temperature under all three scenarios combined. Darker blue patches represent increasingly ice-free conditions, while whiter patches indicate years and temperatures where a high percentage of emulator ensemble members project high ice cover above 1 million square kilometers. The right panel represents the cumulative probability of the global mean temperature at which the SIA falls below 1 million $km^2$ for the first time in our observationally constrained parameterisations. Red, yellow and blue represent SSP5-8.5, SSP2-4.5 and SSP1-2.6 respectively. Errorbars and scattered shapes represent probability ranges from other studies.

Our analysis indicates that, at the lower bound, we have surpassed the remaining carbon budget by $-14$ Gt of $CO_2$, implying that sufficient carbon has already been emitted to cause a seasonally ice-free Arctic Ocean. However, observational data suggests this is an overestimate as the Arctic Ocean has not yet exhibited seasonally ice-free conditions. We attribute the overestimate to the broader range of our emulator's 2024 SIA projections compared to the more constrained observed values.

Although our emulator considers the full range of observed data, its range is larger than the observed as the 2024 SIA shows very little variation across the available observational datasets. This variance likely arises because our framework is calibrated to the 1850–2100 period over the CMIP6 multi-model range, rather than specifically to the year 2024. Calculating the remaining carbon budget using the same sensitivity and the lower bound of our projected 2024 SIA will produce a smaller budget than the same sensitivity applied to a larger 2024 observed SIA value. This could explain why the lower limit of our emulated

range overestimates the carbon budget although the sensitivities align. If we had calibrated our parameterisations to match the observed SIA in September 2023 (3.8 million $km^2$), our sensitivity range would suggest a remaining 235 GtCO$_2$ - 1737 GtCO$_2$ could be emitted before ice-free conditions occur.

Further assessment of our carbon budgets against IPCC warming targets reveal that the median remaining carbon budget to prevent an ice-free Arctic Ocean in September is larger than the budget for limiting global warming to $1.5°C$ (275 GtCO$_2$),

yet smaller than the budget limiting warming to $2°C$ (1150 GtCO$_2$) (fig. 9) (Friedlingstein et al., 2023; Lamboll et al., 2023). Under a high and medium emission scenario (SSP5-8.5 and SSP2-4.5), our emulator framework projects SIA will *likely* (IPCC likelihood scale defined in Sect. S1.5) become ice-free at $1.9°C$, whereas only 38% of ensembles project an ice-free Arctic Ocean in September at $1.5°C$. Interestingly, the probability of an ice-free ocean under a low emission scenario (SSP1-2.6) significantly increases when constraining the sensitivity. Although the probability does not become *likely* by the IPCC definition

of 66%, the probability peaks at 59% in 2068 before declining as SIA recovers (fig. 9).

## 5.5 Non-linear Regime

Our SASIEv.1 framework suggests that March sea ice declines slowly with $CO_2$ initially, at a rate of $-0.75 m^2/tCO_2$ (IQR: $-0.6$ to $-0.91 m^2/tCO_2$) per emitted ton of $CO_2$ up to a threshold emission of approximately 10,000 GtCO$_2$ from 2024 (IQR: 7156 GtCO$_2$ to 13,596 GtCO$_2$), after which the sensitivity increases to a rate of $-2.2 m^2/tCO_2$ (IQR: $-1.75$ to $-2.93 m^2/tCO_2$)

(fig. 8). The emission threshold we identify corresponds to a median global mean temperature of $5.42°C$ (IQR: $4.349°C$ – $6.62°C$) (fig. 9). Once the emission/ temperature threshold is exceeded, the Arctic ice pack will detach from land year-round, increasing the sensitivity at which sea ice is lost to cumulative carbon emissions. After this threshold has been reached the Arctic Ocean opens up rapidly.

While SIA tends to decline with $CO_2$ emissions at the same rate across all scenarios, rapid ice loss only occurs under the high-

405 emission scenario (SSP5-8.5), while the emission reductions in other scenarios prevent the majority of ensembles reaching the thresholds we identify.

## 6  Discussion

In this study, we present an parameterisation framework that efficiently emulates long-term probabilistic projections of Arctic sea ice that capture current trends and the latest physical understanding of the sea ice system from CMIP6 models. We have demonstrated how our framework uses a comparatively simple set of parameterisations to empirically capture the key features of SIA loss with global warming, compared to the complexity, detailed physics and computational costs of CMIP6 models on which it was trained. Building on this foundation, we show that applying constraints to address the 'hot model' problem and underestimation of Arctic Amplification in CMIP6 models, enables our framework to generate SIA projections that reproduce the 'plausible' (observationally derived) sensitivity well. We highlight that our observational constraint on Arctic Amplification is key to aligning the SASIEv.1 framework's sensitivity with observations, as the sensitivity generated from projections forced by the MAGICC temperature and the CMIP6 calibrated Arctic Amplification do not align with observational data. This suggests that the degree of Arctic amplification in climate models may be key to understanding the future of Arctic sea ice.

Our constrained parameterisations reduce the remaining carbon budget to prevent a seasonally ice-free Arctic Ocean from CMIP6 estimates by 441 $GtCO_2$ from 2024 at its median. This suggests that limiting global warming to 1.5°C (275 $GtCO_2$) is sufficient to prevent a seasonally ice-free Arctic Ocean, whereas 2°C (1150 $GtCO_2$) proves insufficient. These findings indicate greater emission reductions may be required than current policy solutions suggest. While earlier studies have constrained the timing of an ice-free ocean, most recalibrate the modelled Arctic sea ice output, linearly extrapolate observations to ice-free conditions, or select climate models based on their ability to reproduce the observed rate of sea ice loss (Winton, 2011; Niederdrenk and Notz, 2018; Wang et al., 2021; Sigmond et al., 2018; Kim et al., 2023; Jahn et al., 2024; Poltronieri et al., 2024). These methods conclude that ice-free conditions in September are *likely* at 1.8°C with a range between 1.3°C and 2.9°C of global warming (9). Our results agree well with these approaches projecting a *likely* ocean at 1.89°C.

Another crucial insight has been the ability of our emulation framework to capture the rapid loss of winter sea ice to warming outside of the calibration period, from a more extensive ensemble than CMIP6 models are currently able to achieve. We find there is a 50% chance of sea ice detaching from land in March for the emission of a remaining 10,000 $GtCO_2$ from 2024, corresponding to a global mean temperature of 5.42°C (IQR: 3.9°C - 6.3°C). When comparing to previous findings, we project a threshold that is slightly higher than found by Meccia et al. (2020), who use computationally efficient stochastic physics schemes to project rapid winter sea ice loss will occur at $4 \pm 0.35$°C of global warming. Whereas our threshold falls within the lower range from Drijfhout et al. (2015), who used five CMIP5 model simulations (due to the lack of extended runs past 2100) to suggest the threshold occurs at a global mean temperature ranging between 4.5°C and 8.2°C. Our results add to this body of work by providing a more thorough assessment of winter ice loss backed by a wider range of CMIP6 models and observations, which potentially accounts for the slightly different threshold global temperature from our emulation framework, than the two studies discussed here.

While our constraint on Arctic Amplification assumes the ratio will remain constant through the simulation period, some CMIP6 models suggest that the Arctic Amplification will decline as sea ice retreats (Dai et al., 2019; Holland and Landrum, 2021). As the amount of remaining sea ice is usually low or ice-free before it impacts the Arctic Amplification in summer, the feedback mechanism would have minimal influence on our model's projections during this season. Whereas in winter, the rate of sea ice loss after the ice pack detaches from land is so high that changes to the Arctic Amplification past this time have little impact on our SIA projections. While we have shown our constrained Arctic Amplification trend increases the likelihood of an ice-free ocean at lower emission levels compared to CMIP6 estimates, its future is uncertain. It is possible our framework projects a conservative estimate of future Arctic Amplification if current trends continue to persist. If so, the Arctic Ocean could become ice-free earlier than our current projections suggest.

## 7 Conclusions

This study has presented a novel method of probabilistic sea ice projection through the development and application of a parameterisation framework for Arctic sea ice emulation. Our framework's ability to efficiently extend future seasonal SIA based on the present sensitivity to global temperatures while also considering the range of knowledge within CMIP6 models, proves its use as a tool to understand the key gaps in current Arctic sea ice projections outlined in Sect. 1. This method fills the gap in the literature between complex CMIP6 projections, recalibration of model output, and the simple linear extrapolation of sea ice area with anthropogenic forcing. The results from the application of the SASIEv.1 emulation framework bring into question whether mitigation efforts are stringent enough to prevent critical ice loss, while also providing useful information regarding regime changes in sea ice loss that could cause ice-free conditions to occur rapidly year-round.

*Code availability.* The scripts containing the CMIP6 parameterisations, observationally constrained parameterisations, framework setup, application and the scripts used for calibration, are available at https://doi.org/10.5281/zenodo.15252962 (Chilcott et al., 2025).

*Data availability.* The CMIP6 Earth System Model (ESM) output data for both temperature and sea ice area were initially collected from the Earth System Grid Federation (ESGF, 715 https://esgf-node.llnl.gov/projects/cmip6/, last accessed in Jan 2024). The observational temperature datasets were collected from from NASA's Goddard Institute for Space Studies Surface Temperature version 4 (GISTEMP),(Lenssen et al. (2019)), the Berkeley Earth temperature dataset (BEST) (Rohde and Hausfather (2020)) and the Met Office Hadley Centre/Climatic Research Unit version 5.0.1.0 (HadCRUT5) dataset (Morice et al. (2021)). Observational Northern hemispheric SIA datasets were collected from the University of Hamburg (UHH) Sea Ice Area Product (Rauschenbach et al. (2024)). The six products used include: the HadISST NSIDC monthly sea-ice area; HadISST original monthly sea-ice area; Comiso-Bootstrap monthly sea-ice area; NASA-Team monthly sea-ice area; OSI-SAF monthly sea-ice area and the Walsh monthly sea-ice area. Cumulative $CO_2$ emission projections are taken from the historical

fossil fuel and industrial datasets that were used for harmonising IPCC emissions scenarios (IPCC (2022), AR6 WG3; Nicholls et al. (2020), RCMIP datasetes).

*Author contributions.* All authors conceptualised the study. SC conducted data collection, analysis, and validation and served as the primary developer of the framework's parameterisations, with co-development and supervision from MM and DN. SC also performed the model application analyses and prepared the initial manuscript draft, while MM and DN contributed to manuscript review and editing.

*Competing interests.* The contact author has declared that none of the authors has any competing interests.

*Acknowledgements.* We are also deeply grateful to Peter Rayner for his invaluable support and advice throughout this project. SC's work was also funded by the Melbourne Research Scholarship and the Dr. Albert Shimmins Fund. Finally, we extend our thanks to the many individuals whose dedication made the datasets used in this study possible.

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
