# Peer review of "SASIEv.1: A framework for seasonal and multi-centennial Arctic sea ice emulation"

_Geoscientific Model Development, 2024_

## Author Comment (AC1)

We thank the reviewer for their detailed comments, constructive suggestions and appreciate the time taken to review this work. We have carefully considered all comments and have provided our response below.

In the following, reviewer comments start with a **R:** and are set in grey italics, while our responses start with a **A:** and are in red.

**Anonymous Referee #1**

**1.   General: Model Evaluation:**

*R: The model evaluation is done looking at extensions of the datasets out to 2300, from a calibration of the scenarios up to 2100. To me, a more useful model evaluation would consist of applying the framework to other scenarios run by the same model. Most models presented here will have run one or more additional scenarios such as SSP-3.70 or SSP-1.19. In addition, many of them have run multiple ensemble members. Considering how the fits would do in that context to explore variability would make a lot of sense to me, at least producing some plots to see how the parametrization fits in the range of the climate models internal variability.*

A: We thank the reviewer for their interesting thoughts regarding possible emulator evaluation methods. We agree with the reviewer that an understanding of how our emulator captures internal variability is an important one. While we did consider this option along with a number of other evaluation methods, we decided to focus our evaluation on the long-term performance of our Arctic sea ice projections for the following reasons:

The long-term variability expressed across different ensemble members is comparatively small relative to the model to model spread. In addition, internal variability plays less of a role in long-term Arctic sea ice uncertainty than structural uncertainty (Bonan et al, 2020). We therefore chose not to evaluate the performance of our emulator by testing its ability to reproduce other ensemble members, as this might lead to the false impression of a close emulation given the strong similarities across different ensemble members. While internal variability is important, it is largely constrained within a single model's ensemble due to shared physics and parameterisations. Testing against another ensemble member may thus overstate emulator skill, as it would not demonstrate an ability to generalise across broader climate conditions. Instead, by validating against different external forcing scenarios over long time periods, we ensure that the emulator accurately represents the system's long-term trajectory, which is one of the main aims of this paper. Focusing the verification on time periods outside of the calibrated domain allows us to evaluate whether the emulator can faithfully represent both historical and future changes, making it a more robust test of emulator suitability.

In addition, one of the intentions of this study is to understand whether the physical relationship between SIA and temperature between 1850 and 2100 provides information on future sea ice loss. This provided another reason to test whether our emulator could capture the non-linearity of winter sea ice loss outside of the calibration period, as it provides a way to understand whether future sea ice loss can be understood from the physical mechanisms over the calibration period.

Although our emulator does not attempt to capture the internal variability, we have added a figure to the supplementary (Figure 1 below), applying our calibration parameters from ensemble member 1 to other ensemble members for CanESM5, to test their application. This simple test shows our emulator does capture other ensemble members for this model well. In addition, we have also added an evaluation of our CMIP6 parameterisations with SSP scenarios not used in the calibration process (specifically SSP3-7.0), as we agree with the reviewer that this is a valuable form of parameterisation evaluation. We have added this to the supplementary, as the focus of this paper is focused on how the observational constraints to the CMIP6 calibrated parameterisations and long-term projections further our understanding of Arctic sea ice possibilities, rather than an evaluation of our parameterisations.

[Figure]

*Figure 1| Testing our calibrations on the internal variable of non-calibrated ensemble members from the CMIP6 model CanESM5.*

**2. General: Overshoot:**

*R: Another point of particular interest to the applications for this tool which I am missing some discussion of, is overshoot. Can this emulation deal with that, or do you believe it is beyond the validity of the parametrizations? Is it only beyond that if accelerated tipping points occur before the maximal warming is achieved, or will it also be true otherwise. I think some discussion of the applicability of this tool to overshoot scenarios is necessary here.*

**A:** The reviewer suggests an interesting application for our emulator. One of the scenarios we train our emulator parameterisations on is SSP1-2.6, where temperature recovers slightly, leading to a corresponding recovery in our emulated sea ice (Fig. S13). Although this isn't an overshoot scenario it shows that if the temperature recovers, our projections follow the ESMs scenarios and also recover.

However, we don't train our model on actual overshoot scenarios. Some models that simulate overshoot suggest the Arctic and northern latitudes could remain cooler under overshoot scenarios due to the AMOC weakening and the longer time it takes to recover after peak temperatures (Liu et al, 2020; Jackson et al 2015, Schwinger et al, 2022). However, our sea ice projections are forced by temperature, so if the northern hemisphere temperature remained colder for longer, it's reasonable that our sea ice projections would show sea ice growth here. Furthermore, while it is likely scenarios with lower overshoot (small increases above 1.5°C or 2°C) would have little impact on winter sea ice, it also is unclear how scenarios with larger overshoot (larger temperature increases before decline), could affect the non-linearity of winter sea ice.

As our emulator hasn't been trained on such conditions, we agree with the reviewer that it would be interesting to test our emulator on overshoot scenarios. While this is an interesting application, an analysis of overshoot applications using our emulator is outside the scope of this initial analysis. We believe an analysis of overshoot would be better placed in further emulator applications where an in-depth analysis and discussion can be undertaken. We do not believe a deep enough analysis can be discussed here, if included into the revised manuscript. We will add a few sentences to the revised manuscript to mention overshoot and its interest as a further application. Further work should therefore examine how overshoot is captured in our emulator in a first step. While a second step could train our emulator on overshoot

scenarios. This would allow our emulator to be used as a tool to understand Arctic sea ice under overshoot scenarios more robustly.

**3. General: Framing**

*R: When I come into a model description paper, what I hope to see is a description a model that can be reused also by others outside the group of the authors, where the code is nicely reusable, with interesting scientific applications and somewhat of a finished product. In this case (and a lot of other cases), what I see is rather the description of a really interesting parametrization framework for a Arctic temperatures and sea ice, calibrated to model output, but with functional forms and based on empirical and mechanistic expert judgement. I also see a workflow in steps, but it is less clear how the code would allow me to apply that workflow to new datasets. Instead the code looks more like supporting material that would allow the reproduction of the current results, rather than something that is meant for further use (at least by outsiders). (An added problem here is the lack of a proper README-file for the code, and the choice of programming language, which requires licensed software (Matlab) possibly with an unknown number of add-on libraries with their own libraries to be run). All of this takes nothing away from the work presented, it is very nice and merits publication, but I think this is less the description of "an emulator" and more the description of an emulation framework (personally I'm not sure the word emulation is even the best here, as currently it seems to be used to mean practically everything, hence the meaning of it gets sucked out of the term). I will give more advice in detail below, on what would help in making this a better description paper regardless of whether we are describing an emulator or a framework, however, I'd like the authors to consider what they think this article is describing in the way that they write it, as that might also add to the clarity of the presentation.*

**A:** The reviewer makes a very important point regarding the framing of our manuscript. We have labelled the manuscript as a model description paper, however we agree it is more of a framework paper.

Our emulator code also does not currently support reusability. The code is not easily re-usable as it can only be used with specific datasets, as the reviewer mentions, to reproduce the current work. We therefore agree with the reviewer that we have outlined a parameterisation framework, rather than a body of code that can be used on a number of datasets and run easily.

As the purpose of this paper is to understand how the use of a simple model impacts long-term sea ice projections, we therefore intend for this manuscript to showcase an analysis using the described parameterisation collection on questions within the sea ice discourse, rather than a fully-fledged model that can be used on a range of datasets. We intend to update our emulator setup and code in further versions to be an easily reproducible model. Whereas here, we rather intend to highlight our emulator as an initial parameterisation analysis, that can be refined in further versions.

When revising the manuscript, we will ensure that our work is clearly framed as a parameterisation framework aimed at exploring whether the set of parameterisations presented can, in the first instance, capture the CMIP6 response and how this framework can be used to understand and address key biases in within the models. While our parameterisations can be adapted and applied by readers in their own work, our intention is not to present a reusable model in this manuscript. Instead, we provide an initial discussion on the potential of this framework to address current questions.

In reference to the .README and code, we discuss this in further detail under comment 9 below.

**4. Title:**

*R: Starting with the title, I believe that one requirement of GMD model description papers is for model name and version to be included in the title. Is the model called "Arctic Sea Ice Emulator v.1"? If so, prefacing the name with the article "an" seems strange. I note that the github code repository for the code has the descriptive yet somewhat unusable name "Development-and-Application-of-an-Arctic-Sea-Ice-Emulator-*

*Journal-Article-2024", if the name is in fact "Arctic Sea Ice Emulator", that would be a better name even in the repository. For a model description paper, I expect a description of a model that can be used more generally, so the model shouldn't be named and coupled this closely to the paper, but this also feeds into my slightly more general point on what you mean for this article to be about.*

**A:** We agree with the reviewer. The title does not accurately reflect the papers intention or the model description requirements.

We did originally intend the model to be titled "Arctic Sea Ice Emulator v.1", however, as the paper is more of a parameterisation framework paper, a complete restructure of the title is probably necessary. The following updated titles may offer possible revisions:

"An Arctic Sea Ice Parameterisation Framework v.1".

Or, perhaps

"The Development and Application of an Arctic Sea Ice Emulator: A Parameterisation Framework v.1"

We will also update the github repository name, so that that it implies the parameterisations at least may be used more widely.
* * *
**5. Abstract:**

*R: Just a couple of what I assume are typos here:*

*R: Line 3: "we developement" -> "we develop"*

*R: Line 5: "emitted to prevent" -> "emitted while preventing" (or something like that. I think nobody is emitting CO2 to prevent seasonally ice free conditions in the Arctic, and if they are that would be a very bad idea...)*

**A:** We thank the reviewer for picking up on these errors, we will correct them in the revised manuscript. In response to the last comment, we do indeed hope no one is emitting to prevent ice free conditions!
* * *
**6. Introduction:**

*R: Line 39: Not sure "ascertain" is the right word here, I think I would prefer something like "find"*

*R Line 49: "we aim assess" -> "we aim to assess"*

**A:** We agree with the author. We will again fix these grammatical issues in the revised manuscript.
* * *
**7. Arctic Sea Ice Emulator Setup: Model description and data:**

Firstly, the reviewer makes a number of great comments that will improve this section of the manuscript, we thank them. As there are a number of interesting comments regarding the model description and data, we have responded to each of the reviewer's comment separately below.

*R: I would maybe not have a subsection headline for the overview but prefer to have that just immediately, but no big deal.*

A: We agree with the reviewer that the additional subsection is a little unnecessary, we will remove this in the revised manuscript.

*R: When going through the overview I would also like references to the sections in which the various steps are described. I would also have preferred to have a separate section on the data used as this is quite instrumental to the workings of the methodology. Currently, the description of what is used for the CMIP6 models is described in the overview in brief, whereas the observational data is described in the data availability section. Ideally, I think both should be described with some discussion of choices made in a separate subsection here, while how to get those datasets should be described for both data types in the data availability section.*

A: The reviewer makes a very good point that is also highlighted by other reviewers. We did not include a good description of all data used and the reasoning for their use in the text. We agree it would be best to include an additional 'Data Collection and Processing' section which we will add below the 'Model Description Overview' section. We also agree that links to each of the sections within the manuscript we discuss in the model overview should be included, we will add this to the revised manuscript.

*R: Tables of involved models and ensemble members used would be useful too. One option if you think this fits poorly in the main text is putting such a section including model and observation data tables in the supplement.*

A: We also agree that a list of all CMIP6 models used is useful to reference here. We will include a table in the supplement that shows each model used and the respective ensemble, however as we focus only on the first ensemble member, this may not be necessary. We will also add a link to this table in the Data Processing and Collection section.

*R: Overall, to be a functioning model description paper I think each subsection should also mention the code/ scripts that implement the procedure described in the section.*

A: We also agree that from a usability perspective, a link to the code would be ideal, if our code repository represented an easily runnable model. However, as we have changed the approach to focus more on a parameterisation framework, we think this could cause confusion. We will instead update the README to point the reader to the appropriate sections in the manuscript.

**8. Arctic Sea Ice Emulator Setup: workflow schematic**

*R: The workflow figure is very nice, but I'd like some more information explaining it in the caption, including information on what sections to refer to for more information on the various steps and workflows involved.*

A: We realise we did not add a significant explanation to the workflow schematic. We will update and add additional information explaining the schematic, with links to the appropriate sections in the revised manuscript.

**9. Arctic Sea Ice Emulator Setup: code**

We have combined all the reviewers' comments regarding the code runnability / REAMDME/ github repository, and provided our response here as there were a few comments regarding these area throughout the reviewers comments.

*R: I am, however, missing a good connection between the text and the code base. How do I run it? How does it work? Given the right data, can I plug and play the whole workflow, or do I need to do each step separately. Can I run it for a different (set of) models or experiments? All of this information doesn't have to be given in detail in the overview, (especially details on how I run it or how it works), but since I also can't find it in the README in the repository, I don't really know where to look, and some of these questions should be answered in the overview, so I know whether I can expect this to be a runnable tool, or a description of a framework that I could possibly reimplement from the description (that is also fine, but again that speaks to what this is and isn't.)*

*R: In the code availability section, I'd like there to be some more details. I think linking to the github repository (so one can easily see it exists) is useful, and some explanations on what requirements to run are (you could also solve that in the code README, at least detailing which Matlab extra libraries are needed), but I think here or somewhere you should let the reader know that your code is in Matlab so they know they might not be able to run it (this is less important if you mainly consider this to be a parametrization scheme see previous comments on this). Regardless, the README should include code flow and how to run to reproduce the various parts of the paper.*

**A:** The reviewers highlight a very important missing link regarding the usability of our code and the in-text discussion. Unfortunately, the whole workflow cannot currently be run from a 'single click', it must be run in separate steps. Given previous comments from the reviewer regarding the intention of the paper, we provide a similar response here, reiterating that as we have written the paper as a parameterisation framework, rather than a complete model that can be taken and used with a number of different datasets, this should be made clear in the paper text and the README. We will update the README in the repository to answer some of these questions, including making clear that we use MATLAB R2024b, and highlight that this is a framework that is intended to be used to reimplement the parameterisations presented. Since we are reframing the paper, we prefer to avoid extensive in-text references to the code, as this is not strictly a model description paper, we will therefore address most of these issues in the README, rather than in-text.

In response to the reviewer's comment regarding the possibility of using our code on other ensembles; while this code is only intended to be used as a method to 'reimplement' the process outlined in the manuscript, where we only use the first ensemble member, it is possible to use our code to calibrate or test other CMIP6 models and ensembles, as it is not solely set up for the models and ensembles we currently use. However, as we don't explicitly use this method in our manuscript, we don't highlight this in the paper text to avoid further confusion of the intention of the paper. However, we will mention in the README that this is possible.

We will address the bulk of these comments in the README rather than the data availability section, as we intend to keep the paper as a framework paper, in response to the reviewers' previous comments. However, we will also provide a brief overview of the usability in the model description to make this clear, as this is important information currently missing from the manuscript.

In regard to the presence of "thesis" within the code, this is an error on our part. This paper is an edited excerpt from my PhD thesis, however it appears we missed some references to the thesis when editing. We apologise for this and will edit some of the code to ensure the Section references are related to the manuscript and not a thesis.

**10. Arctic Sea Ice Emulator Setup: parameterisations**

*R: Another general comment is that there are a lot of parameters and constants defined here, and it seems that the same letters are reused for different things here (though it is not always entirely clear).For instance, is the a in 3c and 3d, the same as the a in 3b? I would much prefer if no letters were reused. Referring early to the parameter tables in the supplement and stating the number of free parameters would be helpful, but also having more subscripts on the parameters to explain what they do, for instance a in 3b could be*

*a_linSIAloss or something like that, that would help in following the equations. I think only a relatively simple going over to clarify this more would help the readability of the equations a lot.*

A: We appreciate the reviewer for pointing these issues out. Updating the parameter names will definitely improve the readability of the manuscript. Firstly, the 'a' in 3c and 3d refers to the same parameter, as 'e' and 'a' are related, we therefore based 'e' on a simple linear regression of 'a'. However, a number of our parameters between other equations e.g the 'a' in the tas(AMAB) equation and the 'a' in the SIA equation are not related, which could definitely cause issues when reimplementing the parameterisations. We will therefore update a number of the variables in each equation and add descriptive subscripts for clarity and to ensure that it is clear which variables relate to each other.

*R: Line 86: The equation for tas(AAAB) given here deserves an equation number and a separate line.*

A: We didn't initially give the Equation for tas(AAAB) on a separate line as we thought the equation was perhaps too simple to warrant a full equation. However, to make it easier to read, we agree with the reviewer that it should be presented in an equation line. We will add this to the revised manuscript.

*R: Line 93: "for the first ensemble member of each", this turn of phrase confuses me here. My understanding was that you are only using the first ensemble member, hence the procedure her is that you find tas(AREF) for whatever model data you are trying to replicate, so in principle, I could choose some other member and do the same?*

A: The reviewer's comment regarding our use of the phrase "for the first ensemble member of each" is interesting. In this section we aimed to clarify that we calculate the absolute Arctic Annual mean temperature using the first ensemble member of each CMIP6 model used as this is the ensemble we calibrate to. However, as we mention this in the model calibration section it perhaps does not need repeating, and is also potentially misleading if the reader applies our framework to other ensemble members. We will therefore remove this section of the text in the revised manuscript.

*R: Line 102: Equation 1, how is this tied to the equation for tas(AAB), is p matched up to fit the linear regression? There are so many parameters and functions floating around, please make it as clear as possible how they fit together.*

A: We provide a more detailed discussion of 'p' and 's' and how they relate to the first equation (tas(AAAB)) in the supplementary, however we agree that reference to this, improvement of our parameter names, and an added few sentences is necessary to make clear what they represent and how they link to other equations in the main text.

*R: Line 126: I know this might be outside your control, but it would be very helpful if this equation came on the next page alongside the next equation and its description*

A: We agree that the positioning of Eq.2a would be ideal above 2b, unfortunately this is not in our control, given the nature of LatTeX. However, given the corrections in the manuscript and addition of a data section below the model overview sections, these are now on the same page closer together in the revised manuscript.

*R: Line 129: "where tas(AAAB)", were you meaning to introduce tas(AMAB) here? That's the new term in the equations? The way it stands now, I got quite confused and had to reread several times...*

*R: Figure 4, caption, second to last line: Duplicate "are" here, "2100 are are displayed"*

*R: Line 161: "which is we define" -> "which we define"*

*R: Line 198, equation 3b: As far as I can see, the term (1 + exp(tas(t=0 -b ))) is a constant (though model and parameter dependent) term. I would consider giving it it's own name and defining it in its own equation, at it makes it easier to read the functional form and evolution of the sea ice area from the equation.*

*R: Line 205: "on the denominator" -> "in the denominator"*

A: Finally, in regard to all other minor corrections mentioned above, we thank the reviewer for highlighting these issues and will correct them in the revised manuscript.

**11. Results**

As several of the reviewer's comments from the general response are discussed further in this 'Results section comments', we refer to our discussion in these initial response sections throughout our responses here.

*R: Line 246-254: A little more detail on this evaluation would be nice. Did you have data out to 2300 for all of the models and experiments considered? Was this part of your model selection criterion? This information could easily have fit into a dataset section in Methods, but if it isn't there, we need that info here (and repetition here might be useful anyway).*

A: We agree with the reviewer that perhaps our explanation is lacking here. We will elaborate on this in an added Data section (discussed in response 7).

*R: All of 3.1: I'd really like a discussion/understanding of why you don't test using other experiments for the same models/ensemble members also. Is it because you would need separate calibration for each experiment? If so that again would make this much less useful as an emulator (but just as useful as a parametrization tool to understand Arctic Sea Ice loss, so no shame in that). Anyway, you should discuss this, discuss whether there could be some extension to other experiments, I'd be particularly interested in overshoot scenarios. Even if you need the scenario itself to get a new parameterization, checking whether the scheme works for overshoot would still be interesting, and I'd be very interested in such an application, or at least hear the arguments why you would think the parametrization might not work for that to understand the limitations of the scheme.*

A: As we discuss our reasoning for our validation approach through our response under the **General: Model Evaluation** section, here I will state briefly again that our parameterisations are flexible. While we have only shown a portion of its ability (as we intended for this paper to briefly showcase an emulator before using it to discuss observational constrain results and how this effects our application), our parameterisations would not require re-calibrating for each ensemble and scenario outside of the calibration scheme used. We calibrated over the 3 extreme scenarios, which allows our parameterisations to capture all experiments in between. We show this in the additional analysis we add to the supplementary where we test our parameterisations ability to capture SSP3-7.0 without calibration to these specific scenarios. We see our emulator captures these well, however as this isn't the focus of our analysis and questioning in this paper,

we put these into the supplementary. We did not make this clear in the original manuscript, but intend to make this clearer in the revised manuscript.

*R: Line 311: From the previous text it looks to me like this should be -14Gt rather than 14Gt*

**A:** We did not initially add a negative sign as we indicate in the sentence that 'we have surpassed the remaining carbon budget by 14Gt', implying the sign is negative. However, we agree with the reviewer that it would improves readability by adding the negative sign in the revised manuscript.

*R: Line 332-337: That is the t in these formulae? tonne of CO2?*

**A:** We thank the reviewer for their comment. The 't' in GtCO2 and $m^2$/t does indicate tonne. We will ensure we make this clearer in the revised manuscript.

**12. Discussion**

*R: Line 367: "who use five" -> "who used five"*

**A:** We thank the reviewer and will correct this in the revised manuscript.

**13. Data availability**

*R: I've commented on this further up, but just noting again that details on the CMIP6 data used and the availability of it should also be listed here.*

**A:** We thank the reviewer for highlighting this distinction between in-text data descriptions and data availability. We have, as discussed earlier, updated the data and data availability section to contain the appropriate information.

**14. Supplement**

*R: Table S1 and table S2, the references in the last sentences should not be "bottom of the table". I assume you are referring to tables S3 and S4 respectively, if so, do that.*

**A:** We agree with the reviewer. We will remove the reference to 'Bottom of the table" from Tables S1 and S2 and place the sentence in the correct tables. We will also move the IPCC definitions to Section S2.3.

*R: Some tables with error scores per model would be good to see.*

**A:** We have provided standard deviations for each of the models and equations used in the supplementary. In the revised thesis we will also provide RMSE values.

*R: S2.2.: This section is strange. Are you just defining the IPCC terms here? If so that doesn't fit the headline very well. To me it looks like this should be a part of Section S2.3, i.e. definitions needed for that.*

**A:** The reviewer is correct, the supplementary section where we define the IPCC range definitions was labelled with an incorrect heading. This section was intended to provide more method and context to Figure 9. We will therefore re-label this header appropriately.

---

## Author Comment (AC2)

We thank the reviewer for their detailed comments, constructive feedback and appreciate the time taken to review this work. We have carefully considered all comments and have provided our response below.

In the following, reviewer comments start with a **R:** and are set in grey italics, while our responses start with a **A:** and are in red.

**Junichi Tsutsui:**

*This paper describes a newly developed Arctic sea ice emulator for use with a more comprehensive climate and carbon cycle emulator such as MAGICC. This sea ice emulator can integrate insights from complex multi-model climate projections combined with observational constraints regarding the Arctic response to changes in global surface temperature. Elaborated parameterizations dealing with key features of Arctic warming patterns and sea ice behaviors result in the successful emulation of complex models' response of the sea ice area to a wide range of warming level pathways over the 21st century and beyond. Emulators can provide a probabilistic approach to the scientific basis for climate change mitigation, and this paper includes such an application study as well. Thus, this paper is suitable for publication in the GMD. However, the current manuscript may require revisions for clarification and consistency within the scope. The following are my concerns and suggestions for consideration.*

1. **L53:**

*R: The acronym SSP has not been described yet. However, it is not necessary to limit the range of scenarios to a specific set of scenarios in this context.*

**A:** We thank the reviewer for catching this, we will define the SSP acronym in the revised manuscript.

The reviewer makes a good point regarding the range of scenarios used. We chose the scenarios presented in this study as they represent a range of high emission and low emission scenarios, while leaving a number of scenarios un-calibrated for testing. Successfully calibrating to a range of scenarios that capture the range of sea ice area response to high emission and low emission warming will ensure our scenarios also capture all warming scenarios in-between. We hope this provides some clarification regarding our choice of scenarios used in this particular study. However, we agree that in future uses of our emulator, it is not necessary to limit the range to SSP scenarios. In the revised manuscript we will remove the 'SSP' acronym from this sentence and define it at the beginning of the Model Description section instead.

2. **Subsection 2.2:**

*R: Are there any drifting errors in the preindustrial control runs for the CMIP6 model variables used in this study? Are these erroneous trends removed?*

**A:** This is a very interesting comment from the author and has provided an interesting discussion for us. As it is largely unknown if PiControl drift errors are a function of global warming level, we have opted to not remove drift here. In addition, as the studies we compare our results with also do not correct for drift, for continuity reasons we also chose not to here. We also test the SIA PiControl which produces negligible drift. However, this has definitely given us food for thought.

3. **Subsections 2.4 and 2.5:**

*R: Given that air temperatures over the sea surface greatly depend on whether the surface is covered by sea ice, decreases in the amplitude of the seasonal cycle, at least partly, reflect decreases in the sea ice area. Is it appropriate to process Steps ii and iii independently?*

*Likewise, is it not necessary to address the temperature biases described in Section 2.4.1, regarding possible sea-ice biases?*

**A:** The reviewer raises an interesting point. We separate Step ii and Step iii primarily to validate each parameterisation independently, making it easier to catch any errors that may occur. While Arctic temperature and sea ice are closely linked, by keeping them distinct, we can systematically address the different biases between models and observations at each stage.

Furthermore, Step ii establishes the seasonality for our SIA parameterisation (Step iii). Forcing SIA with seasonal Arctic temperature allows us to generate seasonal SIA projections. If Steps ii and iii were combined, an additional parameterisation step would be required after the Arctic Amplification parameterisation to reintroduce seasonality. Given this, it is more logical to incorporate seasonality directly through a seasonal temperature amplification.

Our intent is for these parameterisations to be used to understand Arctic sea ice projections and relationships. By separating these steps, we can better investigate the sources of observed trends. Additionally, this separation provides a more flexible emulator, allowing us to analyse Arctic temperature independently and explore a wider range of applications.

Keeping these steps distinct helps refine our understanding of the relationship between seasonal Arctic temperature and sea ice area. Capturing this relationship is crucial, as it enables our emulator to account for the non-linearity of winter SIA beyond the calibration period—a key objective of our study. While the relationships between global temperature and SIA, or Arctic annual temperature and SIA, are well-studied, the connection between Arctic seasonal temperature and seasonal SIA remains less understood. Separating these steps allows our parameterizations to be applied more effectively in addressing this knowledge gap.

In regard to combining the seasonal Arctic temperature and SIA biases, while this could apply to one of the biases, the second temperature bias is essential for the functioning of our observational constraint on SIA. This again helps us understand where the CMIP6 models do not capture observations. Combining the two could incorrectly attribute bias to the SIA, or SIA parameterisation.

While we appreciate the reviewer's valuable feedback, we believe it is more fitting for our framework to handle these steps separately.

4. **Equations (2a), (2b), and (2d):**

*R: It is necessary to clarify the dimensions of the parameters. Although Equations (2a) and (2b) imply that P, f, and h have the dimension of temperature, if so, Equation (2d) makes h dimensionless, which is a contradiction.*

**A:** We thank the reviewer for providing great feedback, clarification of the dimensions will help improve our framework.

As 'h' is a scaling factor that is dependent on temperature, it therefore does not have any units and is dimensionless. 'f' represents the amplitude of our tas(AMAB) function and therefore has temperature units (degC). We realise our equation 2 is a bit misleading in this regard. We will edit equation 3 to show 'h' is dimensionless and is a scaling factor using; tas(AMAB) = P(1+h). We will also add a sentence to the equation description stating that 'h' is a dimensionless function that modifies 'P' and that (2b) is written in a simplified form where 'h' implicitly depends on a dimensioned quantity. We will also make this clear in the

supplementary tables where we provide with the parameter values for each model. This may shine some light on the dimensions of each of the parameters.

**5. L250-251 and Figure 6:**

*R: The acronym RMSE is referred to inappropriately regarding its standard definition, that is, the root mean squared error. The goodness-of-fit statistics are confusing because they are indicated by different indexes: RMSE correlation, correlation coefficient, and RMSE fit.*

**A:** The reviewer highlights an important distinction between the different indexes we use to describe the RMSE. We will correct the goodness of fit statistics in the revised manuscript, to ensure we state we are looking at the root mean square error without the addition of non-standard mathematical indexes. We will also ensure our description of the RMSE is correct.

**6. Figure 6:**

*R: Figure 6 shows relatively large emulation errors for IPSL-CM6A-LR in May. Is there any need to mention anything about this point?*

**A:** We agree with the reviewer, that the quality of fit is slightly worse in May compared to other months. We will add a few sentences of discussion around this, we provide an explanation for this below.

We attribute the poorer fit in May is due to our weighting scheme in the SIA parameterisation. Our approach assumes each month requires the same fraction of the previous month's temperature to capture sea ice loss. However, melting in spring (April–May) requires more energy than ice growth in autumn (September–October), meaning May should have a greater weight. Since our calibration applies a single weight across the seasonal cycle (1850–2100) for all SSPs, the emulator does not account for this seasonal variation. While future versions could refine this with an additional weight, we chose a simpler approach, prioritising the fit in the growth months, which are more relevant for understanding the impacts of seasonal sea ice loss. Despite a poorer fit in the melt season, we find it remains sufficient for our study.

**7. Subsections 3.3 to 3.5 and the second paragraph of Section 4:**

*R: Given that this paper is submitted as a model description paper, the results and discussion in these parts, while providing an interesting application study, give the impression of being out of the scope. The introduction should provide the context of the application use presented in this paper. The related results and discussion should focus on methodological matters rather than mitigation issues, such as the remaining carbon budget, to be consistent with a model description paper.*

**A:** The reviewer makes an important point. We appreciate their feedback regarding the scope of the results and discussion in relation to the paper's classification as a model description paper.

However, our intention is not to present a standalone, runnable model, but rather a parameterisation framework designed to understand Arctic sea ice projections. Specifically, we aim to assess whether our approach can reproduce CMIP6 projections and, using these as a basis, correct sensitivity biases when comparing models to observations.

Given this focus, we believe it is appropriate to illustrate the impact of observational constraints on projections, as this directly informs the utility of our framework. While methodological considerations remain central, the broader discussion of model biases and their implications for sea ice projections is integral to demonstrating the relevance of our parameterisation approach.

We therefore respectfully maintain that including the mitigation issues discussed align with the purpose of the paper and provides necessary context for its application within sea ice discourse. However, we acknowledge the importance of clarity in framing the study and will refine the introduction to better emphasise the intended scope and objectives.

**8. L289-291:**

*R: 'slightly larger' is valid for sensitivity to temperature, but not for sensitivity to CO2 emissions. The right panel in Figure 7 shows 'slightly smaller' for the latter.*

**A:** We agree with the reviewer, they are correct in saying that our observationally constrained emulator projects a sensitivity to global warming that is slightly larger, but a sensitivity to CO2 emissions that is slightly smaller. While we do specify we are discussing the global temperature when we say "slightly larger", we will make it clear in the revised manuscript, that the against CO2 emissions it is "slightly smaller".

**9. Code and Data availability:**

*R: The README document in the code archive should include a description of the programming environment and where to obtain the input data necessary for execution.*

**A:** The reviewer provides valuable feedback, aligning with other reviewer comments.

We will update and improve the README in the repository to clarify the programming language and where to obtain the data. We have cited the observational data we have used, rather than providing a link as these can change regularly, however we realise we did not cite the Earth System Grid Federation (ESGF) as the source of our CMIP6 data. We will also make clear that we use MATLAB R2024b, and highlight that this is a framework that is intended to be used to reimplement the parameterisations presented, rather than a 'single-click' runnable model.

*R: The current manuscript describes what data were used but does not describe how they can be obtained. The observational and reference data used in this study are described in the main text.*

**A:** We have not included a sufficient description of all data used and how they are obtained. To address this, we have included an additional section 'Data Collection and Processing', which we will add below the 'Model Description Overview' section. Here we will discuss in further detail which data is used, from all CMIP6, observational, RCMIP6 and MAGICC sources, while in the Data Availability section we will provide citations for where we obtained the data.

**10. Figure 1:**

*R: The figure caption would be more suitable with 'calibration and constraining processes' than with 'conceptual model.' Font size should be increased for readability.*

**A:** We agree with the reviewer that a better description of the model may be necessary. We will take the reviewers advice and update the figure title/ caption.

In combination to another reviewer's comment, we will add more information to the Figure 1 caption and update the caption to 'Conceptual schematic of the parameterisation framework', or 'A work-flow of the calibration and constraining processes in our parameterisation framework'.

**11. Figure 2:**

*R: Is the description 'Ranges indicate the 17th-83rd percentile of the scenario range' correct?*

*I assume that it is a model-structure uncertainty range rather than a scenario range.*

**A:** The reviewer is correct, we are referring to the 17th-83rd percentile of the structural uncertainty for each of the SSP scenarios presented, rather than the 17th-83rd percentile of the scenario range. We will update our figure caption appropriately.

**12. Figure 3:**

*R: The figure caption should include a description of each panel. What is the histogram at the right end of panel (a)? Panel (a) shows that the values for CMIP6 models fall to zero from place to place. What does this mean?*

**A:** We will take the reviewer's advice on board and update the figure caption for figure 3 to more thoroughly describe the plot. We discuss this further in the supplementary however, the histogram represents the distribution of Arctic Amplification values, highlighting their spread and frequency, as derived from the randomly sampled 's' and 'p' variables.

Additionally, 'Panel (a) shows that the values for CMIP6 models fall to zero from place to place' as the Arctic Amplification represents the warming ratio between the Arctic and global temperature anomalies. The low signal to noise ratio for small warming grades in some models cause the Arctic Amplification to near 0 when global warming anomaly or the Arctic temperature anomaly is still relatively small in a given year. This therefore makes the AA fluctuations more pronounced.

**13. Figure 7:**

*R: If the quantity of CO2 emissions is cumulative, it should be clearly indicated from which point in time it has accumulated.*

*The unit of weight should be clearly stated as tCO2 or tC.*

**A:** The reviewer makes a very good point, we will include the point in time (1750) from which the CO2 emissions have accumulated.

We will also update figure 7 to $m^2/tCO_2$ if it makes the figure clearer, however we initially used $m^2/t$ as this has generally been the standard unit used in other studies discussing the sea ice sensitivity to CO2.

**14. L126:**

*R: In Equation (2a), the outermost and innermost parentheses are redundant.*

**A:** We have removed the outermost parentheses from Equation (2a), however the innermost parentheses are required to group the variables we apply the cosine to. Unless we have misunderstood the reviewer's comment, in which case we would be happy to discuss further.

**15. L496:**

*R: In this journal, references may be provided with full author lists. I notice that the author list of Nicholls et al. (2020) is incomplete.*

**A:** While this is true, for aesthetic reasons we choose to keep the authors in the reference list as short as possible, as we believe the list reference list could become messy and unreadable as I have cited a number of papers with a large number of authors.
* * *
**16. L129:**

*R: Considering the difference in the number of days in each month, the equally spaced m values do not exactly represent each month.*

**A:** The reviewer has highlighted an interesting point. While we agree, we technically emulate the average temperature in each month given by 'm', and our plotting interpolates between these points. However, we agree that our wording in this sentence is misleading. We will correct it to a more appropriate description of 'm'. Perhaps the following could be a possible revision: 'm' is an equally spaced value between 0 and 2pi representing the average point of each month of the year (0 and 2pi represent January of year 't' and January of year 't+1' respectively).

We would be happy to discuss this further with the reviewer if our revision suggestion does not adequately describe 'm'.
* * *
**17. Minor comments and technical corrections:**

**A:** In regard to the following minor comments and technical corrections, we will address all the minor revisions in the revised manuscript, and thank the reviewer for picking up on them.

**Figure labels and markers:**

*R:  The labels in the figures and the median markers in the box plots are too small.*

**L18:**

*R: The acronym SIA is not explicitly defined.*

**L129**

*R: Considering the difference in the number of days in each month, the equally spaced m values do not exactly represent each month.*

**L136**

*R: To avoid 'moves ... vertically' that implicitly assumes the orientation of a graph plot, how about rewriting 'a non-optimised parameter that moves the temperature curve vertically to ensure ...' to 'a non-optimised offset value to ensure ...'?*

*Likewise, 'vertically shifted' on L207 should be rewritten appropriately.*

**L180:**

*R: The description of the seasonal asymmetry of SIA changes may need some literature.*

**L285**

*R: Probably, 'than' is an error for 'that.'*

**L289**

*R: Delete 'm' just after '-2.5'*

**Figure 4**

*R: The caption includes duplicated 'are'*

**Figures 4 and 5**

*R: Although the selected CMIP6 models are arbitrary examples, it appears preferable to show the same set of models in these figures.*

**Supplementary tables**

*R: The units of the values are not stated.*

---

## Author Comment (AC3)

We thank Juan for their comments and constructive feedback. In the following, reviewer comments start with a **R:** and are set in grey italics, while our responses start with a **A:** and are in red.

**Juan A. Añel**

**Code Usability Clarification in Manuscript:**

*R: I would like to request you to clarify in your manuscript what it is necessary to run your code. I mean, you have provided the code for your model in the M language. This can be run using proprietary software or free software. The ideal case would be that it can be run using free software, such as GNU Octave, to assure the replicability of your work, which the use of proprietary software precludes. Also, beyond identifying the name of the software/interpreter compatible with your code, please, provide the exact version number used to run it, and if possible all the versions of the software compatible with your code.*

**A:** We thank Juan for their valuable comments and feedback. As this is a comment all three reviewers have made, we realise, and agree that it is important to appropriately clarify the usability of our emulator code in both our manuscript and an updated README.

We agree that ensuring the code can run on free software is an important step toward improving accessibility and reproducibility. However, for this study, our primary goal was to present a series of parameterisations to enhance the understanding of sea ice loss rather than to provide a fully runnable model. (We realise from other reviewers' comments that we have not made this clear in the current manuscript, that our intention for this paper to represent a parameterisation framework rather than an easily runnable model. We will make this clear in the revised manuscript). At this stage, the emulator was developed using MATLAB (R2024b), and we have not yet tested its compatibility with free alternatives such as GNU Octave or Python modules. We acknowledge the importance of making future versions of the emulator freely accessible and plan to work toward this in subsequent releases. However, we believe this process would be too time consuming for this current manuscript. In the meantime, we will clarify the software requirements within the manuscript and provide the exact version number used to ensure transparency.

---

## Author Response (AR1)

We again thank all reviewers for their detailed comments, constructive suggestions and appreciate the time taken to review this work. We have carefully considered all comments and have provided our response and stated any changes to the manuscript below.

In the following, reviewer comments start with a **R:** and are set in grey italics, while our responses start with a **A:** and are in red, while any changes to the manuscript are outlined under **AC:.**

Response and revisions to comments from 'Anonymous Referee #1' are presented first, followed by comments from Junichi Tsutsui on page 11.

**Anonymous Referee #1**

**1.    General - Model Evaluation:**

*R: The model evaluation is done looking at extensions of the datasets out to 2300, from a calibration of the scenarios up to 2100. To me, a more useful model evaluation would consist of applying the framework to other scenarios run by the same model. Most models presented here will have run one or more additional scenarios such as SSP-3.70 or SSP-1.19. In addition, many of them have run multiple ensemble members. Considering how the fits would do in that context to explore variability would make a lot of sense to me, at least producing some plots to see how the parametrization fits in the range of the climate models internal variability.*

**A:** We thank the reviewer for their interesting thoughts regarding possible emulator evaluation methods. We agree with the reviewer that an understanding of how our emulator captures internal variability is an important one. While we did consider this option along with a number of other evaluation methods, we decided to focus our evaluation on the long-term performance of our Arctic sea ice projections for the following reasons:

The long-term variability expressed across different ensemble members is comparatively small relative to the model to model spread. In addition, internal variability plays less of a role in long-term Arctic sea ice uncertainty than structural uncertainty (Bonan et al, 2020). We therefore chose not to evaluate the performance of our emulator by testing its ability to reproduce other ensemble members, as this might lead to the false impression of a close emulation given the strong similarities across different ensemble members. While internal variability is important, it is largely constrained within a single model's ensemble due to shared physics and parameterisations. Testing against another ensemble member may thus overstate emulator skill, as it would not demonstrate an ability to generalise across broader climate conditions. Instead, by validating against different external forcing scenarios over long time periods, we ensure that the emulator accurately represents the system's long-term trajectory, which is one of the main aims of this paper. Focusing the verification on time periods outside of the calibrated domain allows us to evaluate whether the emulator can faithfully represent both historical and future changes, making it a more robust test of emulator suitability.

In addition, one of the intentions of this study is to understand whether the physical relationship between SIA and temperature between 1850 and 2100 provides information on future sea ice loss. This provided another reason to test whether our emulator could capture the non-linearity of winter sea ice loss outside of the calibration period, as it provides a way to understand whether future sea ice loss can be understood from the physical mechanisms over the calibration period.

**AC**: While we intend to evaluate the performance of our emulator by assessing its ability to capture long term Arctic sea ice using the information from calibration between 1850 and 2100, we have added two figures to the supplementary (**Fig. S11 & S12**). The first applies our calibration parameters from ensemble member 1 to other ensemble members for CanESM5, to test their application. This simple test shows our emulator does capture other ensemble members for this model well. In addition, we have also added an evaluation of our CMIP6 parameterisations with SSP scenarios not used in the calibration process (specifically SSP3-7.0), as we agree with the reviewer that this is a valuable form of parameterisation

evaluation. We have added this to the supplementary, as the focus of this paper is focused on how the observational constraints to the CMIP6 calibrated parameterisations and long-term projections further our understanding of Arctic sea ice possibilities, rather than an evaluation of our parameterisations.

**2.    General - Overshoot:**

*R: Another point of particular interest to the applications for this tool which I am missing some discussion of, is overshoot. Can this emulation deal with that, or do you believe it is beyond the validity of the parametrizations? Is it only beyond that if accelerated tipping points occur before the maximal warming is achieved, or will it also be true otherwise. I think some discussion of the applicability of this tool to overshoot scenarios is necessary here.*

**A:** The reviewer suggests an interesting application for our emulator. One of the scenarios we train our emulator parameterisations on is SSP1-2.6, where temperature recovers slightly, leading to a corresponding recovery in our emulated sea ice (Fig. S13). Although this isn't an overshoot scenario it shows that if the temperature recovers, our projections follow the ESMs scenarios and also recover.

However, we don't train our model on actual overshoot scenarios. Some models that simulate overshoot suggest the Arctic and northern latitudes could remain cooler under overshoot scenarios due to the AMOC weakening and the longer time it takes to recover after peak temperatures (Liu et al, 2020; Jackson et al 2015, Schwinger et al, 2022). However, our sea ice projections are forced by temperature, so if the northern hemisphere temperature remained colder for longer, it's reasonable that our sea ice projections would show sea ice growth here. Furthermore, while it is likely scenarios with lower overshoot (small increases above 1.5℃ or 2℃) would have little impact on winter sea ice, it also is unclear how scenarios with larger overshoot (larger temperature increases before decline), could affect the non-linearity of winter sea ice.

**AC:** As our emulator hasn't been trained on such conditions, we agree with the reviewer that it would be interesting to test our emulator on overshoot scenarios. While this is an interesting application, an analysis of overshoot applications using our emulator is outside the scope of this initial analysis. We believe an analysis of overshoot would be better placed in further emulator applications where an in-depth analysis and discussion can be undertaken. We do not believe a deep enough analysis can be discussed here, if included into the revised manuscript. We have added a few sentences to the revised manuscript to mention overshoot and its interest as a further application (**L324**). Further work should therefore examine how overshoot is captured in our emulator in a first step. While a second step could train our emulator on overshoot scenarios. This would allow our emulator to be used as a tool to understand Arctic sea ice under overshoot scenarios more robustly.

**3.    General - Framing**

*R: When I come into a model description paper, what I hope to see is a description a model that can be reused also by others outside the group of the authors, where the code is nicely reusable, with interesting scientific applications and somewhat of a finished product. In this case (and a lot of other cases), what I see is rather the description of a really interesting parametrization framework for a Arctic temperatures and sea ice, calibrated to model output, but with functional forms and based on empirical and mechanistic expert judgement. I also see a workflow in steps, but it is less clear how the code would allow me to apply that workflow to new datasets. Instead the code looks more like supporting material that would allow the reproduction of the current results, rather than something that is meant for further use (at least by outsiders). (An added problem here is the lack of a proper README-file for the code, and the choice of programming language, which requires licensed software (Matlab) possibly with an unknown number of add-on libraries with their own libraries to be run). All of this takes nothing away from the work presented, it is very nice and merits publication, but I think this is less the description of "an emulator" and more the description of an emulation framework (personally I'm not sure the word emulation is even the best here, as currently it seems to be used to mean practically everything, hence the meaning of it gets sucked out of the term). I will give more advice in detail below, on what would help in making this a better description paper*

*regardless of whether we are describing an emulator or a framework, however, I'd like the authors to consider what they think this article is describing in the way that they write it, as that might also add to the clarity of the presentation.*

**A:** The reviewer makes a very important point regarding the framing of our manuscript. We have labelled the manuscript as a model description, however we agree we describe more of a framework.

Our emulator code also does not currently support easy reusability. Although the code is not easily re-usable as a 'single-click' or 'off the shelf' model, as the reviewer mentions, we have provided the calibration routines and the in-text parameterisations described are intended to be reused not just to reproduce the current work but also can be applied to retrain on other models not trained on in this study. However, we agree with the reviewer that we have outlined a parameterisation framework, rather than a body of code that can be used on a number of datasets and run easily.

As the purpose of this paper is to understand how the use of a simple model impacts long-term sea ice projections, we therefore intend for this manuscript to showcase an analysis using the described parameterisation collection on questions within the sea ice discourse, rather than a fully-fledged model that can be used on a range of datasets. We intend to update our emulator setup and code in further versions to be an easily reproducible model. Whereas here, we rather intend to highlight our emulator as an initial parameterisation analysis, that can be refined in further versions.

**AC:** In the revised manuscript, we have aimed to ensure that our work is clearly framed as a parameterisation framework aimed at exploring whether the set of parameterisations presented can, in the first instance, capture the CMIP6 response and how this framework can be used to understand and address key biases in within the models. While our parameterisations can be adapted and applied by readers in their own work, we have stated that our intention is not to present a reusable or 'off the shelf' model in this manuscript. Instead, we provide an initial discussion on the potential of this framework to address current questions.

In reference to the .README and code, we discuss this in further detail under comment 9 below.

**4. Title:**

*R: Starting with the title, I believe that one requirement of GMD model description papers is for model name and version to be included in the title. Is the model called "Arctic Sea Ice Emulator v.1"? If so, prefacing the name with the article "an" seems strange. I note that the github code repository for the code has the descriptive yet somewhat unusable name "Development-and-Application-of-an-Arctic-Sea-Ice-Emulator-Journal-Article-2024", if the name is in fact "Arctic Sea Ice Emulator", that would be a better name even in the repository. For a model description paper, I expect a description of a model that can be used more generally, so the model shouldn't be named and coupled this closely to the paper, but this also feeds into my slightly more general point on what you mean for this article to be about.*

**A:** We agree with the reviewer. The title does not accurately reflect the papers intention or the model description requirements.

We did originally intend the model to be titled "Arctic Sea Ice Emulator v.1", however, as the paper is more of a parameterisation framework paper, a complete restructure of the title is probably necessary.

**AC:** We have updated the title to the following in the revised manuscript: "SASIEv.1: A framework for seasonal and multi-centennial Arctic sea ice emulation". We have also updated the github repository name, so that that it implies the parameterisations at least may be used more widely.

**5. Abstract:**

*R: Just a couple of what I assume are typos here:*

*R: Line 3: "we developement" -> "we develop"*

*R: Line 6: "emitted to prevent" -> "emitted while preventing" (or something like that. I think nobody is emitting CO2 to prevent seasonally ice free conditions in the Arctic, and if they are that would be a very bad idea…)*

**A & AC:** We thank the reviewer for picking up on these errors, we have corrected them in the revised manuscript (**L3 and L6**).

**6. Introduction:**

*R: Line 39: Not sure "ascertain" is the right word here, I think I would prefer something like "find"*

*R Line 49: "we aim assess" -> "we aim to assess"*

**A & AC:** We agree with the author. We have again fixed these grammatical issues in the revised manuscript (**L39 and 49**).

**7. Arctic Sea Ice Emulator Setup - Model description and data:**

Firstly, the reviewer makes a number of great comments that will improve this section of the manuscript, we thank them. As there are a number of interesting comments regarding the model description and data, we have responded to each of the reviewer's comment separately below.

*R: I would maybe not have a subsection headline for the overview but prefer to have that just immediately, but no big deal.*

**A & AC:** We agree with the reviewer that the additional subsection is a little unnecessary, we have removed this in the revised manuscript.

*R: When going through the overview I would also like references to the sections in which the various steps are described. I would also have preferred to have a separate section on the data used as this is quite instrumental to the workings of the methodology. Currently, the description of what is used for the CMIP6 models is described in the overview in brief, whereas the observational data is described in the data availability section. Ideally, I think both should be described with some discussion of choices made in a separate subsection here, while how to get those datasets should be described for both data types in the data availability section.*

**A & AC:** The reviewer makes a very good point that is also highlighted by other reviewers. We did not include a good description of all data used and the reasoning for their use in the text. We agree it is best to include an additional 'Data Collection and Processing' section which we have added below the 'Model Description Overview' section in the methodology (**L71, Section 3**). We also agree that links to each of the sections within the manuscript we discuss in the model overview should be included, we have added this to the revised manuscript (**L61-70**).

*R: Tables of involved models and ensemble members used would be useful too. One option if you think this fits poorly in the main text is putting such a section including model and observation data tables in the supplement.*

**A &. AC:** We also agree that a list of all CMIP6 models used is useful to reference here. We have included a table in the supplement that shows each model used and the respective ensemble. We have also added a link to this table in the Data Processing and Collection section (**Table. S1, pg37**).

*R: Overall, to be a functioning model description paper I think each subsection should also mention the code/ scripts that implement the procedure described in the section.*

**A & AC:** We also agree that from a usability perspective, a link to the code would be ideal, if our code repository represented an easily runnable model. However, as we have changed the approach to focus more on a parameterisation framework, we think this could cause confusion. We have instead updated the .README to point the reader to the appropriate sections in the manuscript.

**8. Arctic Sea Ice Emulator Setup - workflow schematic**

*R: The workflow figure is very nice, but I'd like some more information explaining it in the caption, including information on what sections to refer to for more information on the various steps and workflows involved.*

**A & AC:** We realise we did not add a significant explanation to the workflow schematic. We have updated the description of the workflow and added additional information explaining the schematic, with links to the appropriate sections in the revised manuscript (**Figure 1, pg4**).

**9. Arctic Sea Ice Emulator Setup: Code**

We have combined all the reviewers' comments regarding the code runnability / REAMDME/ github repository, and provided our response here as there were a few comments regarding these area throughout the reviewers comments.

*R: I am, however, missing a good connection between the text and the code base. How do I run it? How does it work? Given the right data, can I plug and play the whole workflow, or do I need to do each step separately. Can I run it for a different (set of) models or experiments? All of this information doesn't have to be given in detail in the overview, (especially details on how I run it or how it works), but since I also can't find it in the README in the repository, I don't really know where to look, and some of these questions should be answered in the overview, so I know whether I can expect this to be a runnable tool, or a description of a framework that I could possibly reimplement from the description (that is also fine, but again that speaks to what this is and isn't.)*

*R: In the code availability section, I'd like there to be some more details. I think linking to the github repository (so one can easily see it exists) is useful, and some explanations on what requirements to run are (you could also solve that in the code README, at least detailing which Matlab extra libraries are needed), but I think here or somewhere you should let the reader know that your code is in Matlab so they know they might not be able to run it (this is less important if you mainly consider this to be a parametrization scheme see previous comments on this). Regardless, the README should include code flow and how to run to reproduce the various parts of the paper.*

**A:** The reviewers highlight a very important missing link regarding the usability of our code and the in-text discussion. Unfortunately, the whole workflow cannot currently be run from a 'single click', it must be run in separate steps. Given previous comments from the reviewer regarding the intention of the paper, we provide a similar response here, reiterating that as we agree that we have written the paper as a

parameterisation framework, rather than a complete model that can be taken and used with a number of different datasets, this should be made clear in the paper text and the .README.

**AC:** We have updated the .README in the repository to answer some of these questions, including making clear that we use MATLAB R2024b, and highlight that this is a framework that is intended to be used to reimplement the parameterisations presented. Since we are reframing the paper, we prefer to avoid extensive in-text references to the code, we therefore address most of these issues in the README, rather than in-text. However, we are happy to discuss this further with the reviewer's if this isn't sufficient.

**AR**: In response to the reviewer's comment regarding the possibility of using our code on other ensembles; this code is intended to be used as a method to both 'reimplement' the process outlined in the manuscript, and also run the framework on other CMIP6 scenarios and ensembles. While we only discuss the 13 models we calibrate to in this study, it is possible to use the code to retrain other models, and is not necessary to retrain or recalibrate the code for other scenarios than those used. For example, the code can be run on any of the SSP scenarios for any of the 13 models calibrated to without needing to re-calibrate the parameterisations, however the code would need to be recalibrated for application on other models, we provide this code in the Zenodo repository under 'calibration'. While not a 'one click runnable tool', the code can be used for other possibilities outside those described in the paper. Although somewhat reusable, we have not tested the code on scenarios outside of the SSP ESM scenarios calibrated to (e.g overshoot scenarios).

**AC:** We have mentioned both in-text and in the README that it is possible to recalibrate or retrain the presented parameterisations on models outside of those used in calibration, and encourage this, we also clarify that the parameterisations can be used on scenarios and ensemble members outside of calibration for the models used in the study, using the calibration parameters provided without the need for recalibration (**L59 & 60**).

We have addressed the bulk of these comments in the .README rather than the 'Data Collection and Processing' or 'Framework Overview' sections, as we intend to keep the paper as a framework paper, in response to the reviewers' previous comments. However, we do also provide a brief overview of the usability in the 'Framework Overview' (**L59 & 60**) to make this clear, as this is important information currently missing from the manuscript.

In regard to the presence of "thesis" within the code, this is an error on our part. This paper is an edited excerpt from my PhD thesis, however it appears we missed some references to the thesis when editing. We apologise for this and have edited some of the code to ensure the Section references are related to the manuscript and not a thesis.

**10. Arctic Sea Ice Emulator Setup: parameterisations**

*R: Another general comment is that there are a lot of parameters and constants defined here, and it seems that the same letters are reused for different things here (though it is not always entirely clear).For instance, is the a in 3c and 3d, the same as the a in 3b? I would much prefer if no letters were reused. Referring early to the parameter tables in the supplement and stating the number of free parameters would be helpful, but also having more subscripts on the parameters to explain what they do, for instance a in 3b could be a_linSIAloss or something like that, that would help in following the equations. I think only a relatively simple going over to clarify this more would help the readability of the equations a lot.*

**A:** We appreciate the reviewer for pointing these issues out. Updating the parameter names will definitely improve the readability of the manuscript. Firstly, the 'a' in 3c and 3d refers to the same parameter, as 'e' and 'a' are related, we therefore based 'e' on a simple linear regression of 'a'. However, a number of our

parameters between other equations e.g the 'a' in the tas(AMAB) equation and the 'a' in the SIA equation are not related, which could definitely cause issues when reimplementing the parameterisations.

**AC:** We have therefore updated a number of the variables in each equation and add descriptive subscripts for clarity and to ensure that it is clear which variables relate to each other.

*R: Line 116: The equation for tas(AAAB) given here deserves an equation number and a separate line.*

**A & AC:** We didn't initially give the Equation for tas(AAAB) on a separate line as we thought the equation was perhaps too simple to warrant a full equation. However, to make it easier to read, we agree with the reviewer that it should be presented in an equation line. We have added this to the revised manuscript (**L116, eq. 1**).

*R: Line 121: "for the first ensemble member of each", this turn of phrase confuses me here. My understanding was that you are only using the first ensemble member, hence the procedure her is that you find tas(AREF) for whatever model data you are trying to replicate, so in principle, I could choose some other member and do the same?*

**A:** The reviewer's comment regarding our use of the phrase "for the first ensemble member of each" is interesting. In this section we aimed to clarify that we calculate the absolute Arctic Annual mean temperature using the first ensemble member of each CMIP6 model used as this is the ensemble we calibrate to. However, as we mention this in the model calibration section it perhaps does not need repeating, and is also potentially misleading if the reader applies our framework to other ensemble members. We have therefore removed this section of the text in the revised manuscript (**L121**).

*R: Line 130: Equation 1, how is this tied to the equation for tas(AAB), is p matched up to fit the linear regression? There are so many parameters and functions floating around, please make it as clear as possible how they fit together.*

**A:** We agree with the reviewer that reference to p and s (which have now been updated to a more readable parameter name), improvement of the parameter names, and an added few sentences is necessary to make clear what p & s represent and how they link to other equations in the main text. We have addressed these issues in Section 4.2.1.

*R: Line 157: I know this might be outside your control, but it would be very helpful if this equation came on the next page alongside the next equation and its description*

**A:** We agree that the positioning of Eq.2a (now Eq.3q) would be ideal above 2b (now 3b), unfortunately this is not in our control, given the nature of LatTeX. However, given the corrections in the manuscript and addition of a data section below the model overview sections, these are now on the same page closer together in the revised manuscript.

*R: Line 160: "where tas(AAAB)", were you meaning to introduce tas(AMAB) here? That's the new term in the equations? The way it stands now, I got quite confused and had to reread several times…*

*R: Figure 4, caption, second to last line: Duplicate "are" here, "2100 are are displayed"*

*R: Line 194: "which is we define" -> "which we define"*

*R: Line 230, equation 4b: As far as I can see, the term (1 + exp(tas(t=0 -b ))) is a constant (though model and parameter dependent) term. I would consider giving it it's own name and defining it in its own equation, at it makes it easier to read the functional form and evolution of the sea ice area from the equation.*

*R: Line 238: "on the denominator" -> "in the denominator"*

A: Finally, in regard to all other minor corrections mentioned above, we thank the reviewer for highlighting these issues and have corrected them in the revised manuscript. (The revisions can be found on the lines indicated above, we have updated them so they are relevant to the revised manuscript).
* * *
**11. Results**

As several of the reviewer's comments from the general response are discussed further in this 'Results section comments', we refer to our discussion in these initial response sections throughout our responses here.

*R: Line 275-280: A little more detail on this evaluation would be nice. Did you have data out to 2300 for all of the models and experiments considered? Was this part of your model selection criterion? This information could easily have fit into a dataset section in Methods, but if it isn't there, we need that info here (and repetition here might be useful anyway).*

A: We agree with the reviewer that perhaps our explanation is lacking here. We have elaborated on this in an added 'Data Collection and Processing' (discussed in response 7).

*R: All of 5.1: I'd really like a discussion/understanding of why you don't test using other experiments for the same models/ensemble members also. Is it because you would need separate calibration for each experiment? If so that again would make this much less useful as an emulator (but just as useful as a parametrization tool to understand Arctic Sea Ice loss, so no shame in that). Anyway, you should discuss this, discuss whether there could be some extension to other experiments, I'd be particularly interested in overshoot scenarios. Even if you need the scenario itself to get a new parameterization, checking whether the scheme works for overshoot would still be interesting, and I'd be very interested in such an application, or at least hear the arguments why you would think the parametrization might not work for that to understand the limitations of the scheme.*

A: As we discuss our reasoning for our validation approach through our response under the **General: Model Evaluation** section, here I will state briefly again that our parameterisations are flexible. While we have only shown a portion of its ability (as we intended for this paper to briefly showcase an emulator before using it to discuss observational constrain results and how this effects our application), our parameterisations would not require re-calibrating for each ensemble and scenario of the models used. We calibrated over the 3 extreme scenarios, which allows our parameterisations to capture all experiments in between.

AC: We have demonstrated this in additional analysis where we tested our parameterisations ability to capture SSP4-6.0 without calibration to these specific scenarios. We see our emulator captures these well, however as this isn't the focus of our analysis and questioning in this paper, we put these into the supplementary (**Fig. S12**). We did not make this clear in the original manuscript, however we have added a

few sentences to the revised manuscript to hopefully provide more of an understanding why we evaluated our parameterisations by evaluating their ability to reprodice long-term sea ice (**L314-329**).

*R: Line 366: From the previous text it looks to me like this should be -14Gt rather than 14Gt*

**A:** We did not initially add a negative sign as we indicate in the sentence that 'we have surpassed the remaining carbon budget by 14Gt', implying the sign is negative. However, we agree with the reviewer that it would improves readability by adding the negative sign in the revised manuscript (**L366**).

*R: Line 361: That is the t in these formulae? tonne of CO2?*

**A:** We thank the reviewer for their comment. The 't' in GtCO2 and $m^2/t$ does indicate tonne. We have corrected this in the revised manuscript to hopefully clarify exactly the units we are referring to. We have labelled the '$m^2/t$' as '$m^2/tCO_2$', and also defined 'GtCO2' as Gigatonnes of CO2 before using its shortened unit form.

**12. Discussion**

*R: Line 423: "who use five" -> "who used five"*

**A:** We thank the reviewer and have corrected this in the revised manuscript (**L423**).

**13. Data availability**

*R: I've commented on this further up, but just noting again that details on the CMIP6 data used and the availability of it should also be listed here.*

**A:** We thank the reviewer for highlighting this distinction between in-text data descriptions and data availability. We have, as discussed earlier, updated the 'Data Collection and Processing' section to contain the appropriate information.

**14. Supplement**

*R: Table S1 and table S2, the references in the last sentences should not be "bottom of the table". I assume you are referring to tables S3 and S4 respectively, if so, do that.*

**A:** We agree with the reviewer. We have removed the reference to 'Bottom of the table" from Tables S1 and S2 and placed the sentence in the correct tables. We have also moved the IPCC definitions to Section S2.3.

*R: Some tables with error scores per model would be good to see.*

**A:** We have provided standard deviations for each of the models and equations used in the supplementary. In the revised thesis we have also provide RMSE value tables (**Tables S6 & S7**).

*R: S2.2.: This section is strange. Are you just defining the IPCC terms here? If so that doesn't fit the headline very well. To me it looks like this should be a part of Section S2.3, i.e. definitions needed for that.*

**A:** The reviewer is correct, the supplementary section where we define the IPCC range definitions was labelled with an incorrect heading. This section was intended to provide more method and context to Figure 9. We have therefore re-labeled this header appropriately (**S1.5**).

**Junichi Tsutsui:**

*This paper describes a newly developed Arctic sea ice emulator for use with a more comprehensive climate and carbon cycle emulator such as MAGICC. This sea ice emulator can integrate insights from complex multi-model climate projections combined with observational constraints regarding the Arctic response to changes in global surface temperature. Elaborated parameterizations dealing with key features of Arctic warming patterns and sea ice behaviors result in the successful emulation of complex models' response of the sea ice area to a wide range of warming level pathways over the 21st century and beyond. Emulators can provide a probabilistic approach to the scientific basis for climate change mitigation, and this paper includes such an application study as well. Thus, this paper is suitable for publication in the GMD. However, the current manuscript may require revisions for clarification and consistency within the scope. The following are my concerns and suggestions for consideration.*

**15. Section 2:**

*R: The acronym SSP has not been described yet. However, it is not necessary to limit the range of scenarios to a specific set of scenarios in this context.*

**A:** We thank the reviewer for catching this, we have defined the SSP acronym in the revised manuscript.

**AR & AC:** The reviewer makes a good point regarding the range of scenarios used. We chose the scenarios presented in this study as they represent a range of high emission and low emission scenarios, while leaving a number of scenarios un-calibrated for testing. Successfully calibrating to a range of scenarios that capture the range of sea ice area response to high emission and low emission warming will ensure our scenarios also capture all warming scenarios in-between. We hope this provides some clarification regarding our choice of scenarios used in this particular study. However, we agree that in future uses of our emulator, it is not necessary to limit the range to SSP scenarios. In the revised manuscript we have removed the 'SSP' acronym from this sentence and define it at the beginning of the ' Data Collection and Processing' section instead (**L73**).

**16. Section 4:**

*R: Are there any drifting errors in the preindustrial control runs for the CMIP6 model variables used in this study? Are these erroneous trends removed?*

**A:** This is a very interesting comment from the author and has provided an interesting discussion for us. As it is largely unknown if PiControl drift errors are a function of global warming level, we have opted to not remove drift here. In addition, as the studies we compare our results with also do not correct for drift, for continuity reasons we also chose not to here. We also test the SIA PiControl which produces negligible drift.

**17. Subsections 4.3 and 4.3.1:**

*R: Given that air temperatures over the sea surface greatly depend on whether the surface is covered by sea ice, decreases in the amplitude of the seasonal cycle, at least partly, reflect decreases in the sea ice area. Is it appropriate to process Steps ii and iii independently?*

*Likewise, is it not necessary to address the temperature biases described in Section 4.3.1, regarding possible sea-ice biases?*

**A:** The reviewer raises an interesting point. We separate Step ii and Step iii primarily to validate each parameterisation independently, making it easier to catch any errors that may occur. While Arctic temperature and sea ice are closely linked, by keeping them distinct, we can systematically address the different biases between models and observations at each stage.

Furthermore, Step ii establishes the seasonality for our SIA parameterisation (Step iii). Forcing SIA with seasonal Arctic temperature allows us to generate seasonal SIA projections. If Steps ii and iii were combined, an additional parameterisation step would be required after the Arctic Amplification parameterisation to reintroduce seasonality. Given this, it is more logical to incorporate seasonality directly through a seasonal temperature amplification.

Our intent is for these parameterisations to be used to understand Arctic sea ice projections and relationships. By separating these steps, we can better investigate the sources of observed trends. Additionally, this separation provides a more flexible emulator, allowing us to analyse Arctic temperature independently and explore a wider range of applications.

Keeping these steps distinct helps refine our understanding of the relationship between seasonal Arctic temperature and sea ice area. Capturing this relationship is crucial, as it enables our emulator to account for the non-linearity of winter SIA beyond the calibration period—a key objective of our study. While the relationships between global temperature and SIA, or Arctic annual temperature and SIA, are well-studied, the connection between Arctic seasonal temperature and seasonal SIA remains less understood. Separating these steps allows our parameterizations to be applied more effectively in addressing this knowledge gap.

In regard to combining the seasonal Arctic temperature and SIA biases, while this could apply to one of the biases, the second temperature bias is essential for the functioning of our observational constraint on SIA. This again helps us understand where the CMIP6 models do not capture observations. Combining the two could incorrectly attribute bias to the SIA, or SIA parameterisation.

While we appreciate the reviewer's valuable feedback, we believe it is more fitting for our framework to handle these steps separately.
* * *
**18. Equations (3a), (3b), and (3d):**

*R: It is necessary to clarify the dimensions of the parameters. Although Equations (2a) and (2b) imply that P, f, and h have the dimension of temperature, if so, Equation (2d) makes h dimensionless, which is a contradiction.*

**A:** We thank the reviewer for providing great feedback, clarification of the dimensions has helped improve our framework.

As 'h' is a scaling factor that is dependent on temperature, it therefore does not have any units and is dimensionless. 'f' represents the amplitude of our tas(AMAB) function and therefore has temperature units (degC). We realise our equation 2 is a bit misleading in this regard.

**AC:** We have therefore edited equation 3 to show 'h' is dimensionless and is a scaling factor using; tas(AMAB) = P(1+h) (**L158**). We have also added a sentence to the equation description stating that 'h' is a dimensionless function that modifies 'P' and that (2b) is written in a simplified form where 'h' implicitly depends on a dimensioned quantity (**L166-167**). We have also made this clear in the supplementary tables where we provide with the parameter values for each model. This may shine some light on the dimensions of each of the parameters (**Table S2 & S3, pg. 37 & 38**).
* * *
**19. L283:**
* * *
*R: The acronym RMSE is referred to inappropriately regarding its standard definition, that is, the root mean squared error. The goodness-of-fit statistics are confusing because they are indicated by different indexes: RMSE correlation, correlation coefficient, and RMSE fit.*

**A:** The reviewer highlights an important distinction between the different indexes we use to describe the RMSE.

**AC:** We have corrected the goodness of fit statistics in the revised manuscript, to ensure we state we are looking at the root mean square error without the addition of non-standard mathematical indexes. We have also ensured our description of the RMSE is correct.
* * *
**20. Figure 6:**

*R: Figure 6 shows relatively large emulation errors for IPSL-CM6A-LR in May. Is there any need to mention anything about this point?*

**A:** We agree with the reviewer, that the quality of fit is slightly worse in May compared to other months.

**AC:** We have added a few sentences of discussion around this to Section 5.1. We also provide an experp of this discussion for the reviewer below.

We attribute the poorer fit in May is due to our weighting scheme in the SIA parameterisation. Our approach assumes each month requires the same fraction of the previous month's temperature to capture sea ice loss. However, melting in spring (April–May) requires more energy than ice growth in autumn (September–October), meaning May should have a greater weight. Since our calibration applies a single weight across the seasonal cycle (1850–2100) for all SSPs, the emulator does not account for this seasonal variation. While future versions could refine this with an additional weight, we chose a simpler approach, prioritising the fit in the growth months, which are more relevant for understanding the impacts of seasonal sea ice loss. Despite a poorer fit in the melt season, we find it remains sufficient for our study.
* * *
**21. General:**

*R: Given that this paper is submitted as a model description paper, the results and discussion in these parts, while providing an interesting application study, give the impression of being out of the scope. The introduction should provide the context of the application use presented in this paper. The related results and discussion should focus on methodological matters rather than mitigation issues, such as the remaining carbon budget, to be consistent with a model description paper.*

**A:** The reviewer makes an important point. We appreciate their feedback regarding the scope of the results and discussion in relation to the paper's classification as a model description paper.

However, our intention is not to present a standalone, runnable model, but rather a parameterisation framework designed to understand Arctic sea ice projections. Specifically, we aim to assess whether our approach can reproduce CMIP6 projections and, using these as a basis, correct sensitivity biases when comparing models to observations. We also aimed to understand whether information from CMIP6 models between 1850 and 2100 could provide insight into Arctic sea ice in later centuries.

Given these focuses, we believe it is appropriate to illustrate the impact of observational constraints on projections, as this directly informs the utility of our framework. While methodological considerations remain central, the broader discussion of model biases and their implications for sea ice projections is integral to demonstrating the relevance of our parameterisation approach.

We therefore respectfully maintain that including the mitigation issues discussed align with the purpose of the paper and provides necessary context for its application within sea ice discourse. However, we acknowledge the importance of clarity in framing the study and have refined the framing to better emphasise the intended scope and objectives.

**22. L343:**

*R: 'slightly larger' is valid for sensitivity to temperature, but not for sensitivity to CO2 emissions. The right panel in Figure 7 shows 'slightly smaller' for the latter.*

**A:** We agree with the reviewer, they are correct in saying that our observationally constrained emulator projects a sensitivity to global warming that is slightly larger, but a sensitivity to CO2 emissions that is slightly smaller.

**AC:** While we do specify we are discussing the global temperature when we say "slightly larger", we have made it clear in the revised manuscript, that against CO2 emissions it is "slightly smaller".

**23. Code and Data availability:**

*R: The README document in the code archive should include a description of the programming environment and where to obtain the input data necessary for execution.*

**A:** The reviewer provides valuable feedback, aligning with other reviewer comments.

**AC:** We have updated and improved the .README in the repository to clarify the programming language and where to obtain the data. We have cited the observational data we have used, rather than providing a link as these can change regularly, however we realise we did not cite the Earth System Grid Federation (ESGF) as the source of our CMIP6 data. We have therefore also made it clear that we use MATLAB R2024b, and highlight that this is a framework that is intended to be used to reimplement the parameterisations presented, rather than a 'single-click' runnable model (**also L68**).

*R: The current manuscript describes what data were used but does not describe how they can be obtained. The observational and reference data used in this study are described in the main text.*

**A & AC:** We did not include a sufficient description of all data used and how they are obtained. To address this, we have included an additional section 'Data Collection and Processing', which we have added below the 'Model Description Overview' section (**L71**). Here we discuss in further detail which data is used, from all CMIP6, observational, RCMIP6 and MAGICC sources, while in the Data Availability section we provide citations for where we obtained the data.

**24. Figure 1:**

*R: The figure caption would be more suitable with 'calibration and constraining processes' than with 'conceptual model.' Font size should be increased for readability.*

**A:** We agree with the reviewer that a better description of the model may be necessary.

**AC:** We have taken the reviewers advice and updated the figure title/ caption. In combination with another reviewer's comment, we have added more information to the caption of Figure 1 and updated the caption to 'A work-flow of the parameterisation framework'.

**25. Figure 2:**

*R: Is the description 'Ranges indicate the 17th-83rd percentile of the scenario range' correct?*

*I assume that it is a model-structure uncertainty range rather than a scenario range.*

**A:** The reviewer is correct, we are referring to the 17th-83rd percentile of the structural uncertainty for each of the SSP scenarios presented, rather than the 17th-83rd percentile of the scenario range.

**AC:** We have now updated the figure caption appropriately.
* * *
**26. Figure 3:**

*R: The figure caption should include a description of each panel. What is the histogram at the right end of panel (a)? Panel (a) shows that the values for CMIP6 models fall to zero from place to place. What does this mean?*

**A & AC:** We have taken the reviewer's advice on board and updated the figure caption for figure 3 to more thoroughly describe the plot. Here we clarify that the histogram represents the distribution of Arctic Amplification values, highlighting their spread and frequency, as derived from the randomly sampled 's' and 'p' variables.

Additionally, 'Panel (a) shows that the values for CMIP6 models fall to zero from place to place' as the Arctic Amplification represents the warming ratio between the Arctic and global temperature anomalies. The low signal to noise ratio for small warming grades in some models cause the Arctic Amplification to near 0 when global warming anomaly or the Arctic temperature anomaly is still relatively small in a given year. This therefore makes the AA fluctuations more pronounced.
* * *
**27. Figure 7:**

*R: If the quantity of $CO_2$ emissions is cumulative, it should be clearly indicated from which point in time it has accumulated.*

*The unit of weight should be clearly stated as $tCO_2$ or tC.*

**A & AC:** The reviewer makes a very good point, we have included the point in time (1750) from which the $CO_2$ emissions have accumulated.

**AC:** We have also update figure 7 to $m^2/tCO_2$ to ensure the figure is readable and clear, however we initially used $m^2/t$ as this has generally been the standard unit used in other studies discussing the sea ice sensitivity to $CO_2$.
* * *
**28. L156:**

*R: In Equation (2a), the outermost and innermost parentheses are redundant.*

**AC:** We have removed the outermost parentheses from Equation (2a), however the innermost parentheses are required to group the variables we apply the cosine to. Unless we have misunderstood the reviewer's comment, in which case we would be happy to discuss further.

**29. L546:**

*R: In this journal, references may be provided with full author lists. I notice that the author list of Nicholls et al. (2020) is incomplete.*

**A:** This is a good catch, we thank the reviewer.

**AC:** We have now added the full list of author names for the Nicholls et al, (2020) paper and checked all other references.

**30. L159:**

*R: Considering the difference in the number of days in each month, the equally spaced m values do not exactly represent each month.*

**A:** The reviewer has highlighted an interesting point. While we agree, we technically emulate the average temperature in each month given by 'm', and our plotting interpolates between these points. However, we agree that our wording in this sentence is misleading.

**AC:** We have corrected it to a more appropriate description of 'm': 'm' is an equally spaced value between 0 and 2pi representing the average point of each month of the year (0 and 2pi represent January of year 't' and January of year 't+1' respectively). We would be happy to discuss this further with the reviewer if our revision does not adequately describe 'm'.

**31. Minor comments and technical corrections:**

**A:** In regard to the following minor comments and technical corrections, we have addressed all the minor revisions in the revised manuscript, and thank the reviewer for picking up on them.

**Figure labels and markers:**

*R: The labels in the figures and the median markers in the box plots are too small.*

**L19:**

*R: The acronym SIA is not explicitly defined.*

**L166**

*R: To avoid 'moves ... vertically' that implicitly assumes the orientation of a graph plot, how about rewriting 'a non-optimised parameter that moves the temperature curve vertically to ensure ...' to 'a non-optimised offset value to ensure ...'?*

*Likewise, 'vertically shifted' on L139 should be rewritten appropriately.*

**L214:**

*R: The description of the seasonal asymmetry of SIA changes may need some literature.*

**L338**

*R: Probably, 'than' is an error for 'that.'*

**L342**

*R: Delete 'm' just after '-2.5'*

**Figure 4**

*R: The caption includes duplicated 'are'*

**Figures 4 and 5**

*R: Although the selected CMIP6 models are arbitrary examples, it appears preferable to show the same set of models in these figures.*

**Supplementary tables**

*R: The units of the values are not stated.*

---

## Author Response (AR2)

We again thank all reviewers for their comments, constructive suggestions and appreciate the time taken to review this work for a second time. We have again carefully considered all comments and have provided our response and stated any changes to the manuscript below.

In the following, reviewer comments start with a **R:** and are set in grey italics, while our responses and corrections start with a **A:** and are in red**.**

Response and revisions to comments from 'Anonymous Referee #1' are presented first, followed by comments from Junichi Tsutsui.

**Anonymous Referee #1**

We thank the reviewer for carefully identifying the following issues. These have all been addressed in the revised manuscript, and we appreciate the reviewer for picking up on them.
* * *
**R:** *Equation 4a, I suggest swapping w and w-1 out for w_m and w_m-1 (i.e. subscripts). The way it stands now it seems a bit confusing.*
* * *
**R:** *Page 21 line 388: "Further assessment our" -> "Further assessment of our"*
* * *
**R:** *Page 23 line 449: "Our frameworks" -> "Our framework's"*
* * *
**R:** *Page 23 line 459: Are you sure about the last access data for the esgf website? That seems a long time ago for a 2025 article to have checked that..*

**A:** We thank the reviewer for picking on up on this error, this was a typo on our part and is intended to be read as 'Jan 2024'
* * *
**R:** *Page 4, Figure 1 caption: "Refer to sections 4.2.1 and 4.2.1", I suspect one of these duplicate numbers should be something else... 4.2 and 4.2.1?*
* * *
**Junichi Tsutsui:**
* * *
**R:** *To clarify, my concern was primarily about the validity of the cumulative CO2 emissions data. While the temperature data can be directly related to the CMIP6 model output used in this study, the CO2 emissions data are not, which raises questions about the validity and relevance of the results from the sensitivity analysis regarding the remaining carbon budget. Besides, the current manuscript describes the CO2 emissions data from fossil fuels and industrial processes, but not from land use and land-use change. Although some of the CMIP6 models have a carbon-cycle component, evaluating emissions associated with land use are not straightforward.*

**A:** We thank the reviewer for clarifying their comment and apologise for the confusion on our end.

Firstly, we understand that our analysis of SIA against CO2 emissions, the way currently described, may seem a little out of place. While the primary focus of framework development is based in the relation to temperature, we include a sensitivity analysis of the remaining carbon budget to showcase the flexibility of our framework, showing it can be used for analysis on more than just temperature, and also to contextualise our results within broader climate mitigation benchmarks. This addition is intended to show an example of application, rather than extend the scope, while aligning with methods used in prior assessments such as those in the IPCC. We have added a sentence to Section 5.3, pg.19 to clarify this.
* * *
Secondly, in terms of the CO2 emissions data used, we realise we did not update this correctly in the manuscript. Originally, we used only fossil and industrial CO2 emission data however in the submitted iteration of the manuscript, we use fossil, industrial and LULUCF CO2 emission data. We have corrected this in the revised manuscript under the 'Data' description in Section 3, pg. 5.

Thirdly, given the reasons/ discussion above we are eager to keep the carbon budget analysis in the manuscript if possible. We acknowledge that there are differences in the LULUCF (land use, land use change and forestry) emissions between ESMs. However, here we chose a common approach that has also been utilised by the IPCC WG3, where a harmonised LULUCF emissions time series based on historical bookkeeping estimates of anthropogenic emissions are used. We therefore use the same approach to infer cumulative emissions as in IPCC WG3, however we acknowledge that ESMs that infer LULUCF emissions from land-use patterns face uncertainty due to varying model assumptions and the inherent complexity of LULUCF processes, which can lead to differences in the effective LULUCF emissions estimated by different ESMs. We have added another few sentences to Section 3, pg. 5 to hopefully justify our approach and explain the thinking behind the data.

We thank the reviewer for carefully identifying the following minor issues. These have all been addressed in the revised manuscript, and we appreciate the reviewer for picking up on them.

**R:** *Acronym SSP should be defined at its first appearance L68 instead of L75.*

**R:** *The font size used for labels and annotations in Figure 1 appears to be still too small. I recommend increasing the font size to improve the overall clarity and accessibility of the figure.*

**R:** *I agree with the authors' response for equation (3a), not (2a) I believe, but further I have noticed redundant parentheses in the denominator of equation (2).*

**R:** *I understand 'm' represents a reference point for each month. Although I am not confident, 'reference point' may be better wording than 'average point'.*

**R:** *Area units should be km^2, not Km^2, in Table S3.*

**R:** *I have also noticed that the minus sign is represented using a hyphen (-) throughout the text. For clarity and correctness, I recommend using the proper minus sign character (Unicode U+2212) instead.*

---

## Author Response (AR3)

We thank the editor and all reviewers for their feedback and appreciate the time and patience taken to review this work.

We have addressed all the corrections below in the revised manuscript, and we appreciate the editor for picking up on them.
* * *
**R:** *Throughout: Ensure that there is a space between values and units, e.g. 38 Gt instead of 38Gt.*
* * *
**R:** *Line 105: The reference to Nelder and Mead has an extra letter "Nelder and Meadf".*
* * *
**R:** *Code Availability section: Please write cite the Zenodo entry here as a normal citation (i.e., Chilcott (2025)) and place the full reference in the reference list.*
* * *
**R:** *Should the entry for Doerr be (following the guidance on their website):*
*Rauschenbach, Q., Dörr, J., Notz, D., and Kern, S., 2024, UHH sea-ice area product, 1850-2023, University of Hamburg, v2024_fv0.01, https://doi.org/10.25592/uhhfdm.11346, last access date YYYY-MM-DD [fill in date that applies]*
* * *
**R:** *The entries for Balaji et al. (2017) , Hausfather et al. (2022), Poltronieri et al. (2022), Rohde et al. (2020), and Screen et al. (2012) do not have dois.*
* * *
**R:** *The entry for Ritschel 2024 is incomplete.*